# Sulfide catabolism ameliorates hypoxic brain injury

Eizo Marutani[1,2,18], Masanobu Morita[3,18], Shuichi Hirai[1,2,18], Shinichi Kai [1,2], Robert M. H. Grange[1,2], Yusuke Miyazaki[1,2], Fumiaki Nagashima[1,2], Lisa Traeger[1,2], Aurora Magliocca[1,2], Tomoaki Ida[3], Tetsuro Matsunaga[3], Daniel R. Flicker[4,5,6], Benjamin Corman[1,7], Naohiro Mori[1,2], Yumiko Yamazaki[1], Annabelle Batten[1], Rebecca Li [1], Tomohiro Tanaka [8], Takamitsu Ikeda [1,2], Akito Nakagawa[1,2], Dmitriy N. Atochin[2,9], Hideshi Ihara [10], Benjamin A. Olenchock[2,11], Xinggui Shen[12], Motohiro Nishida [8,13], Kenjiro Hanaoka [14], Christopher G. Kevil [12], Ming Xian[15], Donald B. Bloch[1,2,7], Takaaki Akaike[3,19], Allyson G. Hindle [1,2,16,19], Hozumi Motohashi [17,19 ✉] & Fumito Ichinose [1,2,19 ✉]

The mammalian brain is highly vulnerable to oxygen deprivation, yet the mechanism underlying the brain's sensitivity to hypoxia is incompletely understood. Hypoxia induces accumulation of hydrogen sulfide, a gas that inhibits mitochondrial respiration. Here, we show that, in mice, rats, and naturally hypoxia-tolerant ground squirrels, the sensitivity of the brain to hypoxia is inversely related to the levels of sulfide:quinone oxidoreductase (SQOR) and the capacity to catabolize sulfide. Silencing SQOR increased the sensitivity of the brain to hypoxia, whereas neuron-specific SQOR expression prevented hypoxia-induced sulfide accumulation, bioenergetic failure, and ischemic brain injury. Excluding SQOR from mitochondria increased sensitivity to hypoxia not only in the brain but also in heart and liver. Pharmacological scavenging of sulfide maintained mitochondrial respiration in hypoxic neurons and made mice resistant to hypoxia. These results illuminate the critical role of sulfide catabolism in energy homeostasis during hypoxia and identify a therapeutic target for ischemic brain injury.

[1] Anesthesia Center for Critical Care Research of the Department of Anesthesia, Critical Care and Pain Medicine, Massachusetts General Hospital, Boston, MA, USA. [2] Harvard Medical School, Boston, MA, USA. [3] Department of Environmental Medicine and Molecular Toxicology, Tohoku University Graduate School of Medicine, Sendai, Japan. [4] Department of Systems Biology, Harvard Medical School, Boston, MA, USA. [5] Howard Hughes Medical Institute and Department of Molecular Biology, Massachusetts General Hospital, Boston, MA, USA. [6] Broad Institute of MIT and Harvard, Cambridge, MA, USA. [7] Division of Rheumatology, Allergy and Innumology, Department of Medicine, Massachusetts General Hospital, Boston, MA, USA. [8] Division of Cardiocirculatory Signaling, National Institute for Physiological Sciences & Exploratory Research Center on Life and Living Systems & Center for Novel Science Initiatives, National Institutes of Natural Sciences, Okazaki, Japan. [9] Cardiovascular Research Center, Division of Cardiology, Department of Medicine, Harvard Medical School, Massachusetts General Hospital, Charlestown, MA, USA. [10] Department of Biological Science, Graduate School of Science, Osaka Prefecture University, Osaka, Japan. [11] Division of Cardiovascular Medicine, Department of Medicine, The Brigham and Women's Hospital, Boston, MA, USA. [12] Department of Pathology and Translational Pathobiology, Louisiana State University Health Sciences Center-Shreveport, Shreveport, LA, USA. [13] Department of Physiology, Graduate School of Pharmaceutical Sciences, Kyushu University, Fukuoka, Japan. [14] Graduate School of Pharmaceutical Sciences, The University of Tokyo, Bunkyo-ku, Tokyo, Japan. [15] Department of Chemistry, Brown University, Providence, RI, USA. [16] School of Life Sciences, University of Nevada Las Vegas, Las Vegas, NV, USA. [17] Department of Gene Expression Regulation, Institute of Development, Aging and Cancer, Tohoku University, Sendai, Japan. [18] These authors contributed equally: Eizo Marutani, Masanobu Morita, Shuichi Hirai. [19] These authors jointly supervised this work: Takaaki Akaike, Allyson G. Hindle, Hozumi Motohashi, Fumito Ichinose. ✉email: hozumim@med.tohoku.ac.jp; fichinose@mgh.harvard.edu

The brain is the most sensitive organ to hypoxia, presumably due to the brain's high metabolic demand, limited glycolytic capacity, and reliance on oxidative phosphorylation[1–4]. The mechanisms underlying the sensitivity of the brain to hypoxia are incompletely understood. During aerobic respiration, adenosine triphosphate (ATP) is predominantly produced by oxidative phosphorylation using the electrochemical gradient produced via the mitochondrial electron transfer chain (ETC). Molecular oxygen serves as the terminal electron acceptor that binds and accepts electrons from cytochrome c oxidase (COX, complex IV), the last enzyme in the ETC. Approximately 90% of all oxygen consumed by the body is used by COX in a reaction that reduces oxygen in the presence of hydrogen to produce water. When brain tissue $PO_2$ reaches the "critical" level of 6–8 mmHg[5], there is a precipitous drop in cellular ATP concentration resulting in catastrophic energy failure in the brain[6]. Decreased ATP levels in the brain occur within 2–3 min of the onset of severe hypoxia and cause the collapse of the ionic gradient leading to membrane depolarization[5,7]. Persistent lack of oxygen beyond a few minutes irreversibly damages neurons. Paradoxically, the re-introduction of oxygen contributes to further injury[8].

Cytochrome c oxidase has a high affinity for oxygen with a $K_m$ of 0.03–0.3 mmHg, which is significantly lower than the mean $PO_2$ in the brain; the enzyme is saturated with its substrate (i.e., oxygen) under most physiological conditions[5]. Therefore, at the critical brain tissue $PO_2$ level of 6–8 mmHg, when ATP production by oxidative phosphorylation drastically decreases, there appears to be more than sufficient oxygen in the brain tissue for COX to sustain normal electron flow in the ETC. This apparent gap between the critical brain tissue $PO_2$ level and the $K_m$ of COX suggests that there is a role for additional factors that contribute to the inhibition of ETC under hypoxic conditions.

In addition to oxygen shortage, several gases including hydrogen sulfide ($H_2S$), nitric oxide (NO), and carbon monoxide inhibit mitochondrial respiration[9]. $H_2S$ is generally considered a highly toxic substance for aerobic organisms as it inhibits COX[10,11]. However, $H_2S$ also has a number of physiological functions[12]. In mammalian cells, sulfides ($H_2S$ and $HS^-$, which is in equilibrium with $H_2S$), or closely related reactive sulfur species such as persulfides (RSSH) and polysulfides ($RS_nH$), are generated by enzymes including cystathionine β-synthase (CBS), cystathionine γ-lyase (CSE), 3-mercaptopyruvate sulfurtransferase (3-MST), and cysteinyl tRNA synthetase-2 (CARS2)[13–15]. Sulfides are catabolized in mitochondria by sulfide oxidation enzymes. The oxidation of sulfide to persulfide, catalyzed by SQOR, is considered to be the first and rate-limiting step in sulfide oxidation[16]. Persulfide is further oxidized to thiosulfate, sulfite, and sulfate by persulfide dioxygenase (SDO or ETHE1), thiosulfate sulfurtransferase (TST), and sulfite oxidase (SUOX). While SQOR is highly expressed in the liver, heart, lung, skeletal muscle, and colon, the level of SQOR is low in the brains of most mammals including mice, rats, and humans[17] (https://www.proteinatlas.org/ENSG00000137767-SQOR/tissue). Because the brain of these animals has limited capacity to catabolize sulfide[17], the brain is particularly sensitive to the adverse effects of sulfide accumulation. For example, loss of ETHE1 causes fatal sulfide toxicity in ethylmalonic encephalopathy[18]. In addition, the deficiency of SQOR was recently reported to cause Leigh syndrome-like disease characterized by encephalopathy and the presence of brain lesions in the basal ganglia and cortex[19]. These observations suggest a critical role of sulfide catabolism in normal cerebral energy homeostasis. However, the effects of sulfide catabolism on brain bioenergetics during acute oxygen deprivation have thus far attracted little attention.

Under physiological conditions, sulfide oxidation by SQOR donates electrons to mitochondrial ETC complex III via coenzyme Q (CoQ), thereby potentially promoting ATP synthesis[20–22]. In addition, it has been suggested that persulfide produced by SQOR-mediated sulfide oxidation may serve as an electron acceptor from ETC, facilitating mitochondrial ATP production[15,23]. However, hypoxia increases the production of sulfide while inhibiting its oxidation, leading to sulfide accumulation[24–26]. Hypoxia also impedes persulfide oxidation by ETHE1 as this reaction requires oxygen[16]. Excess sulfide promotes the production of NO and reactive oxygen species (ROS)[27–29], which, together with sulfide, impair oxidative phosphorylation during ischemia and increase reperfusion injury[30]. Therefore, sulfide catabolism may play a pivotal role in cerebral energy homeostasis during acute oxygen shortage and in subsequent brain injury upon reoxygenation.

In this study, while examining the effects of chronic intermittent $H_2S$ breathing in mice, we unexpectedly discovered that upregulation of the capacity to catabolize sulfide makes mice remarkably resistant to otherwise lethal hypoxia. Based on this serendipitous discovery, we sought to elucidate the role of sulfide metabolism in the sensitivity of the mammalian brain to hypoxia. We report that increased sulfide oxidation by SQOR makes brains resistant to hypoxia or cerebral ischemia. We also show that pharmacologically scavenging sulfide, so as to avoid sulfide accumulation in the brain, maintains mitochondrial energy production during oxygen shortage and prevents ischemic or hypoxic brain injury. Our study uncovers the critical role of sulfide catabolism in mitochondrial energy homeostasis during hypoxia and lays a foundation for a novel approach to the treatment of ischemic or hypoxic brain injury.

## Results

**Sulfide-pre-conditioning induces hypoxia tolerance in mice.** Breathing $H_2S$ depresses metabolism and decreases the body temperature of rodents[31,32]. However, as we previously showed, intermittent breathing of $H_2S$ for 5 days makes male wild-type mice tolerant to the hypo-metabolic effects of inhaled $H_2S$ (Fig. 1a)[33]. Because chronic sulfide exposure might induce enzymes that increase the capacity to metabolize sulfide, we examined the impact of intermittent $H_2S$ breathing for 5 days (sulfide pre-conditioning, SPC) on sulfide metabolism. Mice were studied 24 h after the last $H_2S$ exposure (on the 6th day), when the metabolism and body temperature of the mice had completely recovered (Supplementary Fig. 1a). When control (not sulfide pre-conditioned) mice acutely breathed $H_2S$, there was a decrease in whole-body metabolism, as measured by a decreased rate of $CO_2$ production ($VCO_2$) (Fig. 1b). In contrast, in mice that were pre-conditioned with $H_2S$, subsequent acute $H_2S$ exposure had no effect on $VCO_2$. Breathing $H_2S$ increased plasma levels of sulfide and thiosulfate to a similar extent in control and sulfide pre-conditioned mice (Fig. 1c, d). However, breathing $H_2S$ increased the levels of sulfide and thiosulfate only in the brains of control mice, but not in the brains of sulfide pre-conditioned mice (Fig. 1e, f). Thus, sulfide pre-conditioning may induce tolerance to the inhibitory effects of $H_2S$ on metabolism by upregulating sulfide catabolism in the brain.

In addition to increased tolerance to breathing $H_2S$, sulfide pre-conditioned mice exhibited marked tolerance to severe hypoxia (5% $O_2$) (Fig. 1g, h and Supplementary Fig. 1b). To examine the impact of sulfide pre-conditioning on biochemical changes in the brain during hypoxia, mice were anesthetized with isoflurane and ventilated with 21% or 5% $O_2$ for 3 min at 37 °C, and brains were harvested and snap-frozen in liquid nitrogen. Breathing 5% $O_2$ decreased brain tissue $PO_2$ below the critical level of 6–8 mmHg within 3 min (Supplementary Fig. 1c). Control mice (not pre-conditioned with $H_2S$) that breathed 5% $O_2$ for 3 min had an

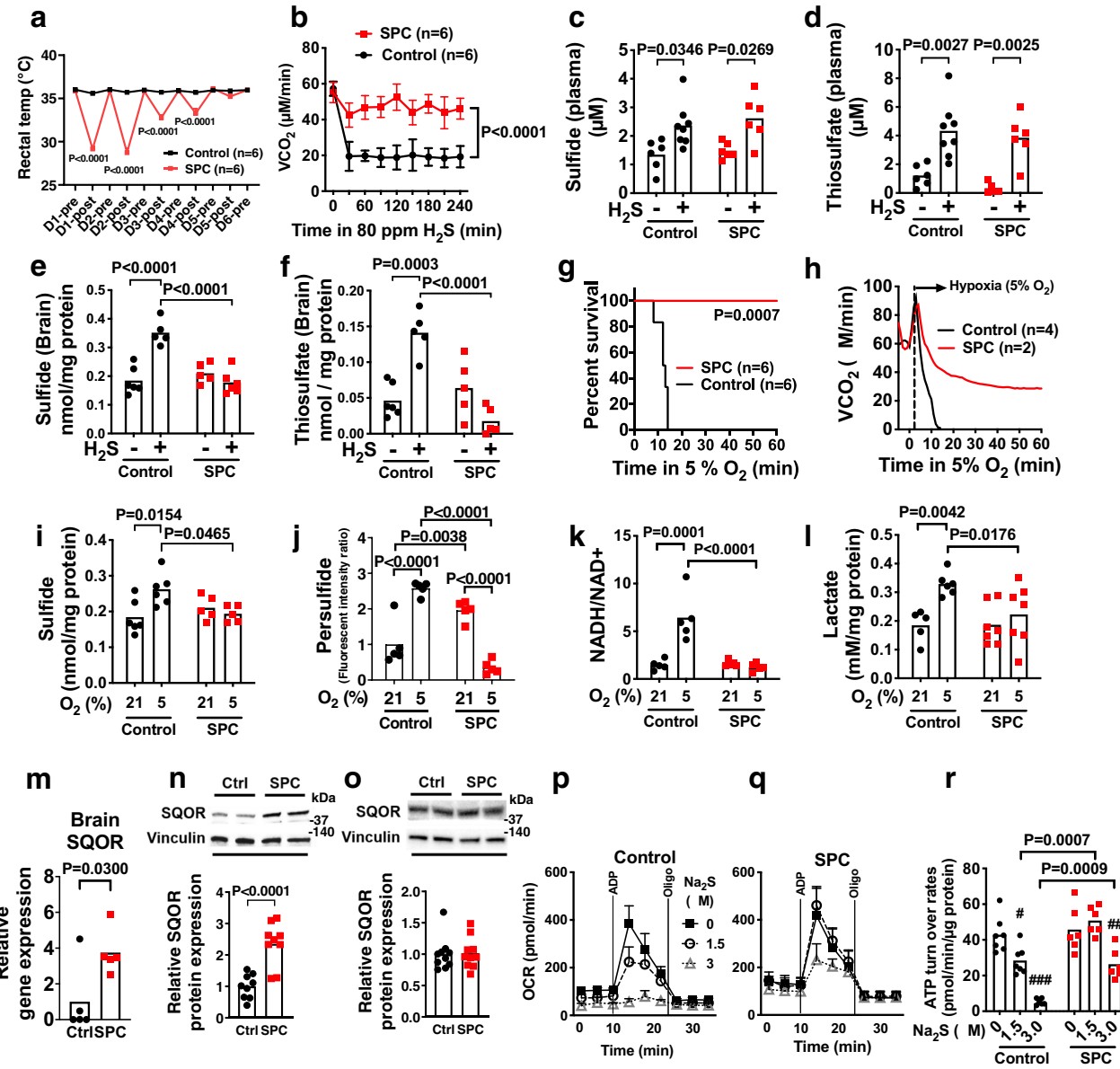

**Fig. 1 Effects of sulfide pre-conditioning on sulfide metabolism and hypoxia tolerance in mice. a** Body temperature of control and sulfide pre-conditioned (SPC) mice. "Pre" and "post" time points depict mice before and after breathing air (control) or $H_2S$ at 80 ppm for 4 h, respectively, from day 1 (D1) to day 6 (D6). **b** Whole-body $CO_2$ production rate ($VCO_2$) of SPC and control mice during $H_2S$ breathing on the 6th day after starting SPC or control air breathing. Concentration of **c** sulfide (from left to right, $n = 6, 8, 6, 6$) and **d** thiosulfate ($n = 6, 8, 6, 6$) in plasma and concentrations of **e** sulfide ($n = 6$ each) and **f** thiosulfate ($n = 6$ each) in the brain of SPC and control mice during $H_2S$ or air breathing on the 6th day. **g** Survival rate and **h** $VCO_2$ of mice breathing 5% $O_2$ on the 6th day in control or SPC mice. Brain levels of **i** sulfide ($n = 6, 6, 5, 5$), **j** persulfide ($n = 5$ each), **k** NADH/NAD$^+$ ratio ($n = 5$ each), and **l** lactate levels ($n = 5, 6, 7, 7$) in control and SPC mice breathing at FiO$_2 = 21\%$ or 5% on the 6th day. **m** Relative mRNA levels of SQOR in the brain of SPC and control mice (normalized as control = 1). $n = 5$ each. SQOR protein levels in **n** the brain tissue extracts ($n = 9$ each) and in **o** the heart ($n = 10$ each) from control and SPC mice. Oxygen consumption rate (OCR) of isolated brain mitochondria from **p** control ($n = 7$) and **q** SPC ($n = 6$) mice with or without incubation with sulfide (Na$_2$S, 0, 1.5, 3.0 μM) and **r** calculated ATP turnover rates ($n = 7, 7, 7, 6, 6, 6$). Data are presented as mean ± SEM or mean and individual values. $^#P < 0.05$, $^{##}P < 0.01$, $^{###}P < 0.001$ vs Na$_2$S at 0 μM of respective group. Two-way ANOVA followed by Sidak's or Tukey's correction for post-hoc comparisons were performed for **a–f**, **i–l**, and **r**. Survival rates were estimated using the Kaplan–Meier method and a log-rank test was used to compare the survival curves between groups in **g**. A two-tailed unpaired $t$-test was performed for **m–o**.

acute increase in brain sulfide levels, to an extent similar to that observed after breathing $H_2S$ at a dose (80 ppm), which inhibits whole-body metabolism (Fig. 1b, e, i). Breathing 5% $O_2$ also increased brain persulfide levels in control mice (Fig. 1j). Increased levels of sulfide in the brain of control mice were accompanied by an increase in the NADH/NAD$^+$ ratio and lactate levels, indicating impaired oxidative phosphorylation (Fig. 1k, l). In contrast, sulfide pre-conditioned mice breathing

5% $O_2$ for 3 min were protected from sulfide accumulation and decreased oxidative phosphorylation in the brain. In addition, brain persulfide levels were decreased in sulfide-pre-conditioned mice breathing 5% $O_2$, suggesting enhanced persulfide consumption during hypoxia (Fig. 1j). Taken together, the results suggest that sulfide pre-conditioning prevents accumulation of sulfide and inhibition of oxidative phosphorylation in the brain of mice after acutely breathing either $H_2S$ at 80 ppm or 5% $O_2$.

To investigate the mechanisms responsible for the inhibition of sulfide accumulation during hypoxia in mice pre-conditioned with $H_2S$, we measured the levels of enzymes that synthesize or catabolize sulfide. Acquired tolerance to acute hypoxia in sulfide pre-conditioned mice was associated with increased levels of SQOR in the brain, but not in the heart presumably due to higher baseline SQOR levels in the heart than in the brain (Fig. 1m–o and Supplementary Fig. 1d). Increments of brain SQOR levels by sulfide pre-conditioning temporarily coincided with the acquisition of hypoxia tolerance; sulfide pre-conditioning for 5 days, but not 2 days, induced hypoxia tolerance and increased brain SQOR levels (Supplementary Fig. 1e, f). Levels of other enzymes that metabolize sulfide were not affected by sulfide pre-conditioning in either brain or the heart (Supplementary Fig. 2a–c). Because sulfide pre-conditioning increased SQOR levels in the brain, we posited that sulfide pre-conditioning upregulates the ability of brain mitochondria to catabolize sulfide and thereby prevents sulfide-induced inhibition of oxidative phosphorylation. Sulfide pre-conditioning did not affect baseline ATP turnover in isolated brain mitochondria as determined by measuring oxygen consumption rates (OCR). In control mice, incubation with $Na_2S$ (a sulfide donor, that mimics hypoxia) dose-dependently decreased ATP turnover in isolated mitochondria obtained from the brain, but not liver that has high basal levels of SQOR (Fig. 1p and Supplementary Fig. 3a–c). In contrast, the ability of $Na_2S$ to depress ATP turnover in the brain mitochondria of sulfide pre-conditioned mice was attenuated (Fig. 1q, r).

Although previous studies suggested that chronic exposure to $H_2S$ increases mitochondrial biogenesis[34], sulfide pre-conditioning did not affect the levels of mitochondrial DNA in the brain or heart of treated mice (Supplementary Fig. 4a). Chronic exposure to hypoxia is known to increase red blood cell mass and the oxygen affinity of hemoglobin[35]. However, five days of sulfide pre-conditioning did not affect either hemoglobin levels or the oxygen dissociation curve of murine red blood cells. Hydrogen sulfide was previously shown to upregulate hypoxia-inducible factor-1α (HIF-1α) and one of its canonical targets vascular endothelial growth factor (VEGF) in vascular endothelial cells and to activate HIF-1α in C. elegans[36,37]. Although sulfide pre-conditioning increased mRNA levels of glucose transporter-1 (GLUT-1), mRNA levels of VEGF, hemoxygenase-1 (HO-1), and erythropoietin (EPO), and protein levels of VEGF, GLUT-1, and lactate dehydrogenase A (LDAH) in the brain did not differ between control and sulfide pre-conditioned mice at 24 h after the last $H_2S$ exposure when mice were exposed to hypoxia (Supplementary Fig. 4b). Taken together, these observations suggest that sulfide pre-conditioning confers tolerance to hypoxia via upregulation of SQOR and sulfide catabolism, rather than as a result of increased mitochondrial biogenesis, increased oxygen delivery by hemoglobin, or upregulation of canonical HIF-1α targets.

**Sexual dimorphism of sulfide catabolism and hypoxia tolerance.** Compared to the brains of 8-week old male CD-1 mice, the brains of age-matched female CD-1 mice have ~3-fold and ~1.5 fold higher levels of SQOR mRNA and protein, respectively (Fig. 2a, b). To investigate the effects of SQOR levels on the ability of the murine brain to tolerate acute oxygen shortage, we subjected CD-1 mice to hypoxia using environmental chambers. The majority of female mice tolerated breathing 5% $O_2$ for at least 1 h, whereas all male mice died in less than 10 min (Fig. 2c). Of note, higher SQOR levels in the brain of female mice appear to be estrogen-dependent. Ovariectomy decreased mRNA and protein levels of SQOR in the brain and abolished hypoxia tolerance in female mice, whereas estrogen supplementation restored brain

SQOR mRNA levels and hypoxia tolerance and tended to restore SQOR protein levels in ovariectomized female mice (Fig. 2d–f).

To investigate the role of SQOR in sulfide catabolism and mitochondrial respiration, we examined the impact of exogenous sulfide, which mimics hypoxia, on mitochondrial respiration in suspensions of brain mitochondria isolated from male and female mice. We used a custom-made spectrophotometer[38] to simultaneously measure the ADP-induced changes in NADH levels, mitochondrial membrane potential (ΔΨm, measured with tetramethylrhodamine, methylester, TMRM), and OCR before and after addition of $Na_2S$. In both male and female mice, in the presence of pyruvate and malate (2.5 mM each), ADP (30 nmol) stimulated oxidative phosphorylation, transiently decreased NADH and ΔΨm, and increased OCR in brain mitochondria (Fig. 2g–i). The addition of $Na_2S$ at 3 or 6 μM (final concentration) inhibited the ADP-induced respiratory stimulation in brain mitochondria from male mice (Fig. 2g, after 3rd ADP addition), but not from female mice (Fig. 2h, after 3rd ADP addition). These observations suggest that higher levels of SQOR in female brain mitochondria confer resistance to hypoxia-induced sulfide accumulation and inhibition of mitochondrial respiration.

To further investigate whether increased brain SQOR levels are responsible for the relative resistance of female mice to hypoxia, we used shRNA targeting mouse Sqor (shSQOR) to knock down SQOR. Adeno-associated virus vectors AAV-shSQOR or control (AAV-Ctrl) were administered at $10^{10}$ viral particles per cerebral hemisphere to newborn female CD-1 mice on postnatal day 0 (P0), via intracerebroventricular (ICV) injection, as described previously (Fig. 2j)[39]. Eight weeks after administration of the AAV, we observed widespread GFP expression in neurons throughout the brain (Fig. 2k, l). Levels of SQOR mRNA, but none of the other enzymes in the transsulfuration and sulfide oxidation pathways, were significantly decreased throughout the brain of mice infected with AAV-shSQOR (Fig. 2m and Supplementary Fig. 5). Compared to 8-week-old control female mice infected with AAV-Ctrl, age-matched female mice with SQOR knockdown were more sensitive to hypoxia induced by exposure to 5% $O_2$ (Fig. 2n). These results support the hypothesis that the higher levels of SQOR in the brains of female mice resulted in a greater capacity to oxidize sulfide and contribute to a greater tolerance to hypoxia.

**Mice that lack SQOR in mitochondria exhibit higher sensitivity to hypoxia.** To elucidate the role of SQOR in mitochondrial bioenergetics during oxygen deprivation, we developed mice that lack SQOR in mitochondria. By using CRISPR-Cas9 technology, we disrupted the translation start codon ATG of the murine Sqor gene via a 14-bp deletion (Fig. 3a). This change led to the initiation of translation starting at a downstream ATG, which resulted in the production of SQOR protein that lacked the N terminal mitochondrial localization sequence (SQORΔN) (Fig. 3c). We confirmed that SQORΔN failed to localize to mitochondria in mouse embryonic fibroblasts obtained from $Sqor^{\Delta N/\Delta N}$ mice (Fig. 3d), although the lack of mitochondrial localization sequence did not affect protein stability of SQORΔN (Supplementary Fig. 6a).

$Sqor^{\Delta N/\Delta N}$ mice were born normally according to the predicted Mendelian ratio. Although $Sqor^{\Delta N/\Delta N}$ mice were indistinguishable from WT littermates before weaning, growth of the homozygous mutant mice ceased around the weaning period (Fig. 3e). All $Sqor^{\Delta N/\Delta N}$ mice gradually became emaciated, developed ataxia, and died within 10 weeks of age, whereas heterozygote mice were normal and similar to WT mice (Fig. 3f, g).

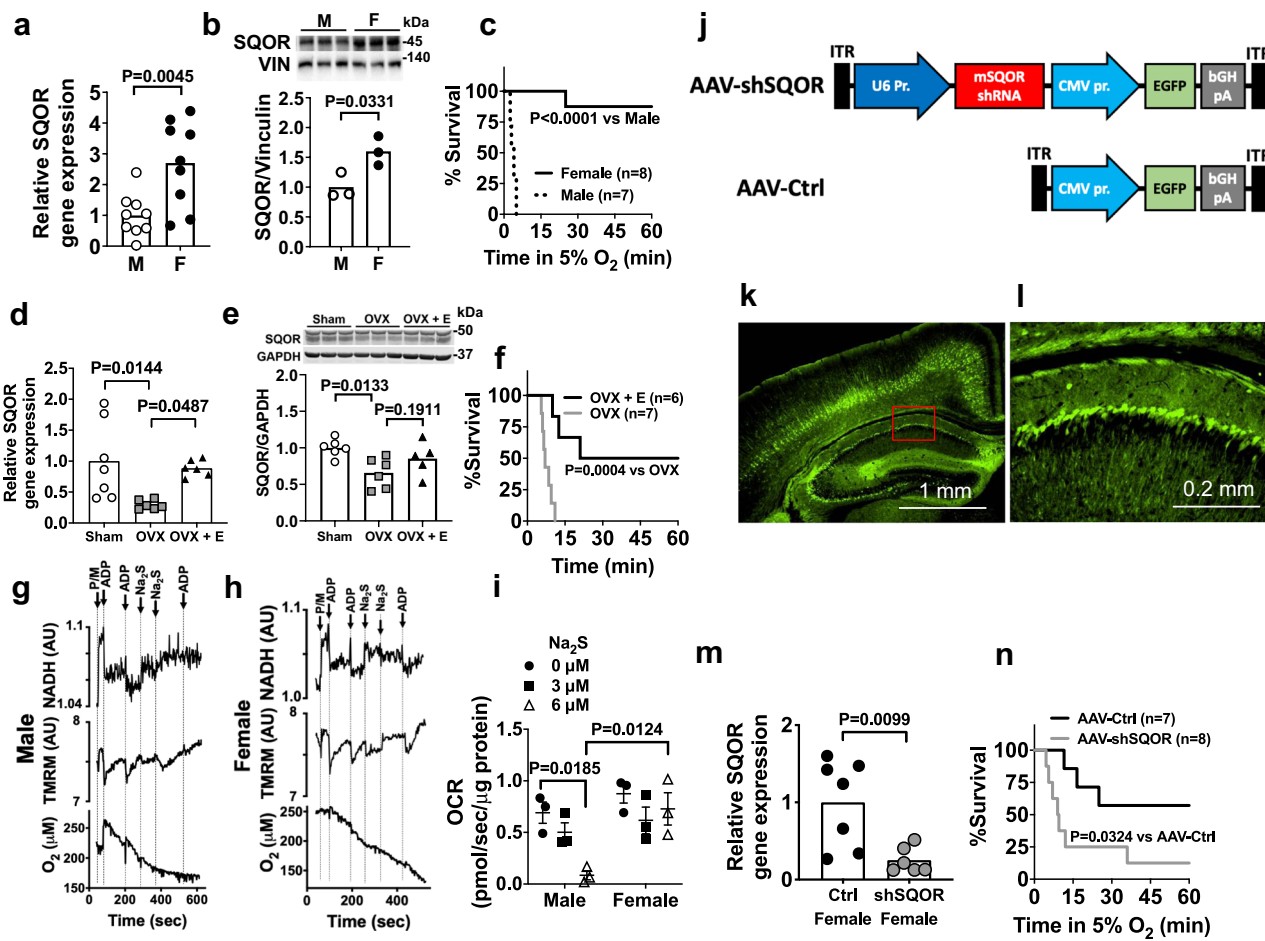

**Fig. 2 Sexual dimorphism of SQOR expression in the brain. a** Relative SQOR mRNA levels ($n = 9$ each) and **b** protein levels ($n = 3$ each) in male and female CD-1 mice. Vin, vinculin. **c** Survival curve of male and female CD-1 mice breathing 5% oxygen. Effects of ovariectomy (OVX) and 17 beta-estradial (E) replacement on **d** SQOR mRNA levels ($n = 7, 6, 6$) and **e** protein levels ($n = 6, 5, 5$) in the brain and **f** survival rate in hypoxia (5% $O_2$) in female CD-1 mice. Impact of sulfide (Na$_2$S) on ADP-induced changes in NADH levels, mitochondrial membrane potential (TMRM, tetramethylrhodamine methylester), and oxygen consumption rates (OCR) in suspensions of isolated brain mitochondria from **g** male and **h** female WT mice. P/M, pyruvate/malate. Representative traces of three independent experiments in each genotype. **i** Summary of OCR values in response to ADP before and after administration of sulfide (Na$_2$S, 0, 3, 6 μM) in experiments represented in **g** and **h**. $n = 3$ at each Na$_2$S dose. **j** Structure of AAV containing shRNA against mouse SQOR (AAV-shSQOR) under a U6 promoter for RNA polymerase III and control AAV (AAV-Ctrl). ITR, inverted terminal repeat, EGFP, enhanced green fluorescent protein, CMV, cytomegalovirus, bGH, bovine growth hormone. **k, l** Representative immunofluorescence images of the brain sections of CD-1 mice stained with an anti-GFP antibody 8 weeks after injection of AAV-Ctrl into ICV. Image in **l** shows a blow-up of a part of the image in **k** enclosed in a red box. $n = 2$ biologically independent experiments. **m** Relative SQOR mRNA levels in the brains of female CD-1 mice transfected with AAV-Ctrl or AAV-shSQOR ($n = 7, 6$). **n** Survival curve of adult female CD-1 mice infected with AAV-Ctrl or AAV-shSQOR breathing 5% oxygen. Data are presented as mean ± SEM or mean and individual values. A two-tailed unpaired $t$-test was performed for **a**, **b**, and **m**. Survival rates were estimated using the Kaplan–Meier method and a log-rank test was used to compare the survival curves between groups in **c**, **f**, and **n**. One-way or two-way ANOVA followed by Dunnett's or Tukey's correction for post-hoc comparisons were performed for **d**, **e**, and **i**.

To examine the role of SQOR in energy homeostasis and survival of neurons in hypoxia, we isolated primary cortical neurons from $Sqor^{\Delta N/\Delta N}$ and WT littermate embryos. $Sqor^{\Delta N/\Delta N}$ neurons had higher levels of intracellular sulfide compared to WT neurons (Fig. 3h), suggesting that SQOR has a critical role in the catabolism of sulfide in neurons even under normoxic conditions. In addition, the magnitude of the oxygen and glucose deprivation (OGD)-induced increase in sulfide levels was markedly greater in $Sqor^{\Delta N/\Delta N}$ compared to WT neurons. Reoxygenation after OGD decreased cell viability more markedly in $Sqor^{\Delta N/\Delta N}$ than in WT neurons (Supplementary Fig. 6b). Compared to 5-week-old WT mice, $Sqor^{\Delta N/\Delta N}$ littermates had a shorter survival when exposed to 5.5% $O_2$ (Fig. 3i). When $Sqor^{\Delta N/\Delta N}$ and control mice breathed air, there were no differences in the levels of sulfide and persulfide or in the ratio of NADH/NAD$^+$ in whole-brain tissue homogenates (Fig. 3j, k and Supplementary Fig. 6c). Although breathing 5.5% $O_2$ for 3 min increased levels of sulfide and persulfide and the ratio of NADH/NAD$^+$ in the brains of both genotypes, sulfide levels and the ratio of NADH/NAD$^+$ were higher in the brains of $Sqor^{\Delta N/\Delta N}$ mice than in control mice. In contrast to the results observed in the brains of WT mice, breathing 5.5% $O_2$ did not affect the levels of sulfide and persulfide or the ratio of NADH/NAD$^+$ in the heart or liver of WT mice (Fig. 3l-o and Supplementary Fig. 6d). However, exposure to 5.5% $O_2$ increased these metrics in the heart and liver of $Sqor^{\Delta N/\Delta N}$ mice. These results indicate that mitochondrial SQOR prevents the accumulation of sulfide and persulfide and decreased oxidative phosphorylation induced by hypoxia not only in the brain but also in the heart and liver.

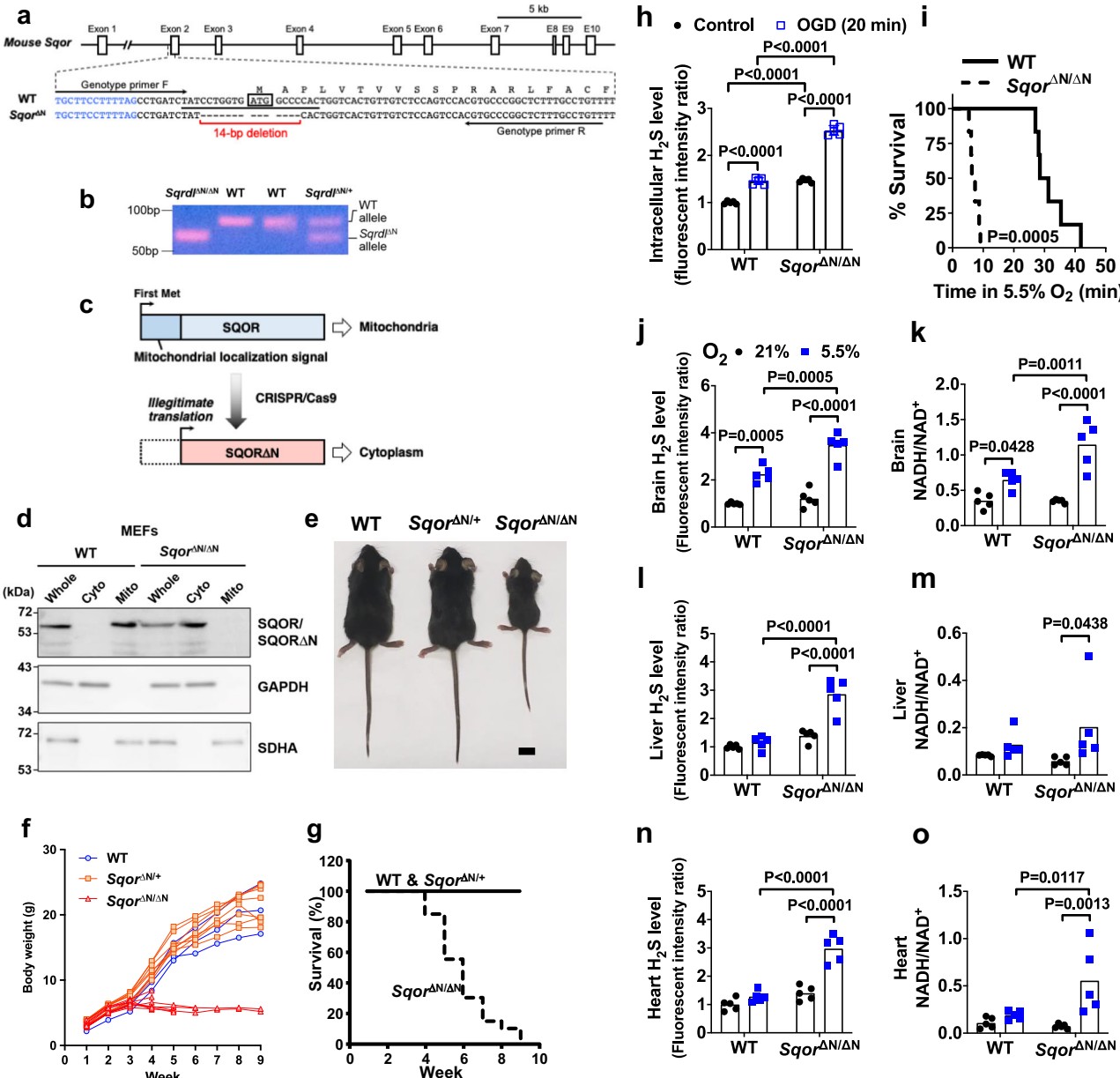

**Fig. 3 Generation of SQOR mutant mice and their increased sensitivity to hypoxia. a** Schematic illustration of the murine *Sqor* gene structure and sequences of WT and mutant alleles spanning the translation initiation codon. Blue letters and black letters indicate the first intron and the second exon, respectively. ATG enclosed by a box indicates the first methionine codon. A target sequence of gRNA is underlined. Target sequences for genotyping primers are indicated by arrows. **b** PCR detection of a deletion in the *Sqor* gene. Genomic DNAs from WT, *Sqor*$^{\Delta N/+}$, and *Sqor*$^{\Delta N/\Delta N}$ mice were amplified with a primer set shown in **a**. Representative PCR gel images of 3 independent biological replicates of each are shown. **c** Schematic presentation of SQOR proteins expressed by WT and *Sqor*$^{\Delta N}$ alleles. **d** Immunoblot analysis of whole-cell lysates (Whole) and cytosolic (Cyto) and mitochondrial (Mito) fractions of MEFs established from WT and *Sqor*$^{\Delta N/\Delta N}$ embryos. Representative immunoblots of 3 independent biological replicates for each are shown. Protein samples were prepared for detection of SQOR, GAPDH (cytosolic marker), and SDHA (mitochondrial marker). **e** Macroscopic appearance of 5-week-old WT, *Sqor*$^{\Delta N/+}$, and *Sqor*$^{\Delta N/\Delta N}$ littermate male mice. Scale bar, 1 cm. Representative images of more than three of each genotype are shown. **f** Body weight gain and **g** survival rate of *Sqor*$^{\Delta N/\Delta N}$ mice (*n* = 20) compared with *Sqor*$^{\Delta N/+}$ mice and WT mice (*n* = 70). **h** Relative intracellular H$_2$S levels of primary cortical neurons obtained and cultured from *Sqor*$^{\Delta N/\Delta N}$ mice and their wild-type littermates embryos subjected to oxygen-glucose deprivation (OGD). **i** Survival curve of *Sqor*$^{\Delta N/\Delta N}$ mice and their wild-type littermates breathing 5.5% oxygen. Relative levels of sulfide in **j**, **l**, **n** and the ratio of NADH/NAD$^+$ in **k**, **m**, **o** in the brain, liver, or heart, respectively, of *Sqor*$^{\Delta N/\Delta N}$ mice and their wild-type littermates breathing 5.5% oxygen (*N* = 5 each). Data are presented as mean and individual values. Two-way ANOVA followed by Tukey's correction for post-hoc comparisons were performed for **h** and **j–l**, **n**, and **o**. Survival rates were estimated using the Kaplan–Meier method and a log-rank test was used to compare the survival curves between groups in **i**. Kruskal–Wallis test followed by Dunn's multiple comparisons was performed for **m**.

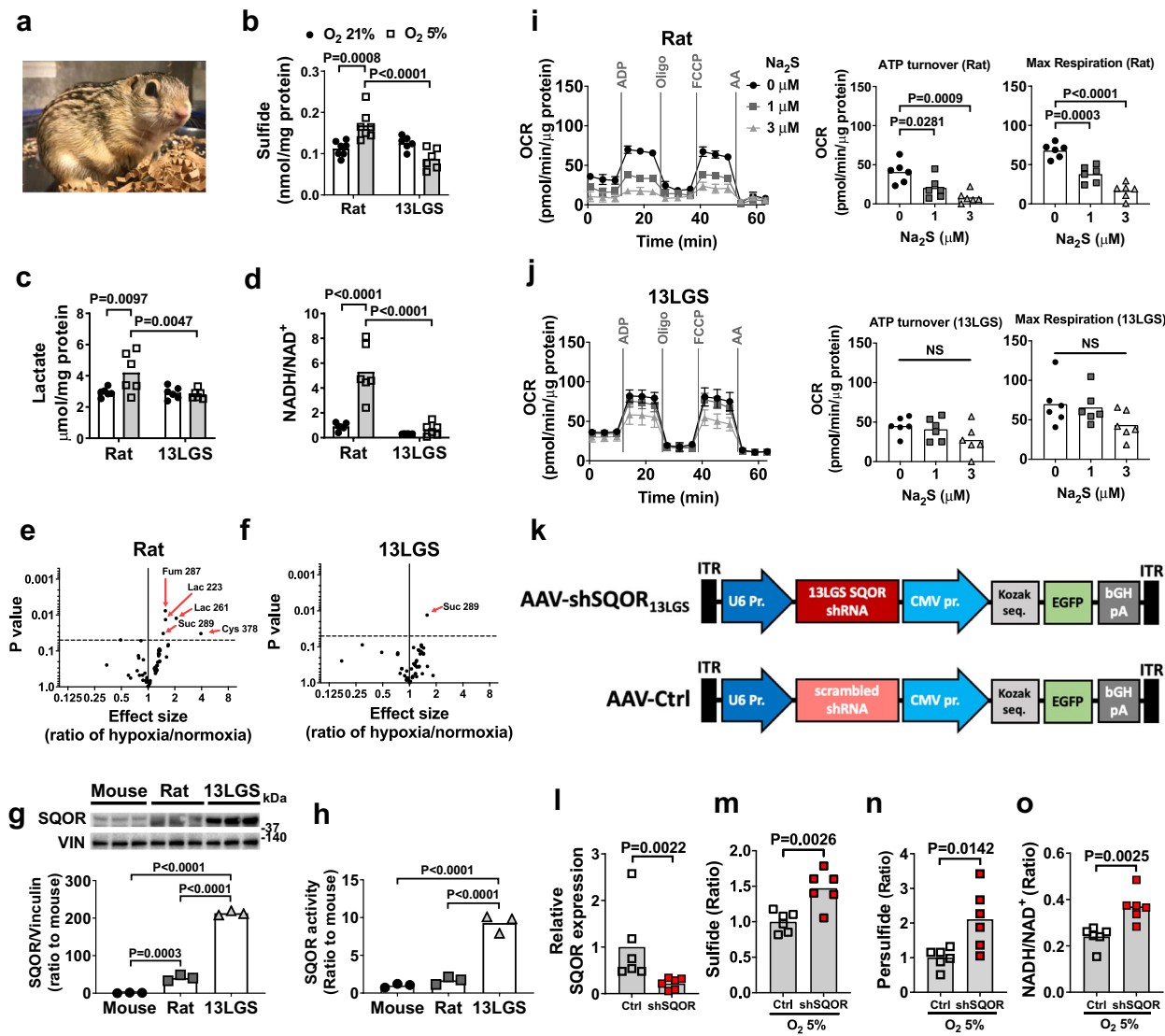

**Fig. 4 Hypoxia tolerance and enhanced sulfide catabolism in 13-lined ground squirrels. a** 13 lined ground squirrel. Levels of **b** sulfide ($n = 7, 7, 6, 6$), **c** lactate ($n = 6$ each), and **d** NADH/NAD$^+$ ratio ($n = 6$ each) in brains of rats and 13LG squirrels (13 LGS) breathing 21% or 5% oxygen for 5 min under isoflurane anesthesia. Volcano plots showing the changes in whole-brain metabolite profiles in response to breathing 5% oxygen in **e** rats and **f** 13LG squirrels. Lac, lactate; Cys, cysteine; Fum, fumarate; Suc, succinate. **g** Immunoblots and protein expression levels of SQOR in forebrains of mouse, rat, and 13LG squirrel ($N = 3$ each). **h** SQOR enzyme activity in forebrains in mouse, rat, and 13LG squirrel ($N = 6$ each). ATP turnover rate and maximal respiratory rate of isolated brain mitochondria measured as oxygen consumption rate (OCR) in **i** rats and **j** 13LG squirrels incubated with 0, 1, or 3 µM of Na$_2$S ($n = 6$ each). **k** Structure of AAV containing shRNA against 13LG squirrel SQOR (AAV-shSQOR$_{13LGS}$) under a U6 promoter for RNA polymerase III and control AAV (AAV-Ctrl) containing scrambled shRNA. **l** Relative SQOR mRNA levels in the brain of 13LG squirrels infected with AAV-Ctrl (Control) or AAV-shSQOR$_{13LGS}$ (shSQOR). Relative levels of **m** sulfide and **n** persulfide and **o** the ratio of NADH/NAD$^+$ in the brains of Control or shSQOR-infected 13LG squirrels breathing 5% oxygen under anesthesia. $n = 6$ each. Data are presented as mean ± SEM or mean and individual values. Two-way or one-way ANOVA followed by Sidak's correction for post-hoc comparisons were performed for **b–d** and **g–j**. Volcano plots were created using values in Supplementary Table S1 and S2. A two-tailed unpaired $t$-test was performed for **l**, **n**, and **o**. Mann–Whitney $U$ test was performed for **m**.

**Hypoxia-tolerant ground squirrels have a high capacity to catabolize sulfide in the brain.** Several mammalian species including 13-lined ground (13LG) squirrels (*Ictidomys tridecemlineatus*) are highly resistant to oxygen deprivation (Fig. 4a)[40–42]. To explore the relationship between sulfide metabolism and natural hypoxia tolerance, we measured brain sulfide levels in hypoxia-resistant 13LG squirrels and hypoxia-sensitive Sprague-Dawley rats. Details of the 13LG squirrel model are provided in the "Methods" section. Brain sulfide levels were comparable between rats and 13LG squirrels breathing room air (Fig. 4b). Breathing 5% O$_2$ for 5 min decreased PaO$_2$ to a similar extent in rats and squirrels (Supplementary Fig. 7a). Levels of

sulfide and lactate and the ratio of NADH to NAD$^+$ were increased in the brains of rats when breathing 5% O$_2$ (Fig. 4b-d). Breathing 5% O$_2$ also increased the levels of homocysteine, but not GSH, thiosulfate, and cysteine, in the brains of rats (Supplementary Fig. 7b). In contrast, exposure to 5% O$_2$ did not affect levels of sulfide, lactate, GSH, homocysteine, thiosulfate, and cysteine and the ratio of NADH to NAD$^+$ in the brains of 13LG squirrels (Fig. 4b-d and Supplementary Fig. 7b). Metabolomic analysis using GC-MS revealed increases in the levels of lactate, fumarate, succinate, and cysteine in the brains of rats following acute hypoxia, while only succinate was increased in the brains of hypoxic 13LG squirrels (Fig. 4e, f and Supplementary Tables 1

and 2). These observations indicate that 13LG squirrel brains do not accumulate sulfide under hypoxia and are resistant to hypoxia-induced impairment of sulfide metabolism and oxidative phosphorylation.

To investigate the mechanisms responsible for the tolerance of 13LG squirrels to the effects of cerebral hypoxia, we measured SQOR levels in the brains of mice, rats, and 13LG squirrels. SQOR protein levels in the brains of 13LG squirrels were ~100-fold greater than that in the brains of mice and rats (Fig. 4g). SQOR enzyme activity, measured as the capacity of brain tissue homogenates to oxidize sulfide using coenzyme $Q_1$ as an electron acceptor[43], was also markedly higher in 13LG squirrels than in mice and rats (Fig. 4h). Protein levels of TST, a sulfide oxidation enzyme downstream of SQOR, were also modestly higher in the brains of 13LG squirrels compared to rats and mice (Supplementary Fig. 7c). In contrast, the abundance and activities of transsulfuration pathway enzymes were comparable between mice, rats, and 13LG squirrels (Supplementary Fig. 7c, d).

To examine the effect of higher SQOR levels and activity on brain mitochondrial function, we measured OCR in isolated brain mitochondria of rats and 13LG squirrels at baseline and in the presence of increasing doses of sulfide (mimicking the effects of hypoxia). Sodium sulfide inhibited oxygen consumption in mitochondria isolated from the brains of rats in a dose-dependent manner. The ability of sodium sulfide to decrease oxygen consumption was attenuated in mitochondria isolated from the brains of 13LG squirrels, indicating a resistance of 13LG squirrel brain mitochondria to the inhibitory effects of sulfide on oxidative phosphorylation (Fig. 4i, j).

To investigate the role of SQOR in the ability of 13LG squirrels to tolerate brain hypoxia, we designed and produced an AAV containing an shRNA targeting 13LG squirrel $Sqor$ (AAV-shSQOR$_{13LGS}$). This shRNA was the best of four candidate sequences at decreasing the level of $Sqor$ in myoblasts isolated from 13LG squirrels (Supplementary Fig. 7e and Supplementary Table 3). We packaged shRNA in the AAV9, which, compared to serotypes AAV2, 4, and 8, exhibited the highest transfection efficiency in 13LG squirrel brains (Supplementary Fig. 7f). We administered AAV-shSQOR$_{13LGS}$ or a control AAV with a scrambled sequence (AAV-Ctrl) into the cerebral ventricles of 13LG squirrels at $2.5 \times 10^{11}$ viral particles per hemisphere (Fig. 4k). Two weeks after AAV infection, brain SQOR mRNA was decreased in squirrels that were infected with AAV-shSQOR$_{13LGS}$ compared to those infected with AAV-Ctrl (Fig. 4l). Control and SQOR knockdown 13LG squirrels were anesthetized and ventilated with 5% $O_2$ for 5 min. Compared to control animals, the brains of SQOR knockdown squirrels had increased levels of sulfide and persulfide (Fig. 4m, n). After exposure to 5% $O_2$, the ratio of NADH/NAD$^+$ was higher in the brains of SQOR knockdown squirrels than in the brains of control squirrels (Fig. 4o). The results show that silencing SQOR results in increased levels of sulfide and persulfide and decreased oxidative phosphorylation in the brain of squirrels in response to hypoxia. While there was some variability in the efficiency of SQOR knockdown among animals, there was a negative linear correlation between SQOR mRNA level and the ratio of NADH/NAD$^+$, whereas there was a positive linear correlation between brain sulfide levels and NADH/NAD$^+$ ratio (Supplementary Fig. 7g). These observations indicate that the higher level of SQOR in the brain mitochondria of 13 LG squirrels contributes to the resistance of these animals to acute hypoxia.

**SQOR facilitates ATP production in neuronal cells**. To examine the effects of increased sulfide oxidation on mitochondrial ETC function, SQOR was expressed in the human neuroblastoma cell line SH-SY5Y. The cells expressed very low levels of SQOR when transfected with a control plasmid (Fig. 5a). Expression of SQOR in SH-SY5Y cells incubated in 21% $O_2$ resulted in increased intracellular concentrations of persulfide, indicating that SQOR expression enhanced sulfide oxidation (Fig. 5b). Expression of SQOR did not affect ATP levels in isolated mitochondria incubated in 21% $O_2$ (Fig. 5c). However, when incubated with Na$_2$S (a sulfide donor) in 21% $O_2$, mitochondria isolated from cells expressing SQOR exhibited markedly increased ATP levels compared to mitochondria isolated from control cells. These observations suggest that sulfide oxidation by SQOR contributes to cellular ATP production under normoxic conditions.

When SQOR-expressing and control SH-SY5Y cells were incubated in 1% $O_2$, SQOR prevented the hypoxia-induced increase in sulfide, lactate, and ROS levels, and prevented the hypoxia-induced increase in the NADH/NAD$^+$ ratio (Fig. 5d–g). In addition, expression of SQOR inhibited hypoxia-induced decreases in ATP production and complex IV activity (Fig. 5h, i). These results indicate that enhanced sulfide oxidation by SQOR expression may prevent sulfide accumulation and the impairment of mitochondrial ATP production during oxygen shortage.

**Neuron-specific SQOR expression improves survival in hypoxia**. To determine whether enhanced sulfide oxidation confers resistance to oxygen deprivation in vivo, we used an AAV encoding mouse $Sqor$ (AAV-SQOR) under the hSYN1 promoter to express SQOR specifically in neurons of CD-1 mice (Fig. 6a). In the brains of CD-1 mice treated with AAV-SQOR, the mRNA and protein levels of SQOR, but none of the other proteins that metabolize sulfide, were increased (Fig. 6b, c and Supplementary Fig. 8a and b). Because male CD-1 mice are more sensitive to hypoxia than female mice (Fig. 3c), male and female mice were treated with different concentrations of oxygen. When an 8-week-old male or female mice breathed 5.5% or 4.5% $O_2$, respectively, mice that expressed SQOR in the brain exhibited better survival rates compared to control mice infected with AAV-GFP (Fig. 6d, e).

Because the effects of SQOR expression on indices of oxidative phosphorylation were more robust in male than in female mice presumably due to the lower (baseline) brain levels of SQOR in males (Fig. 3a, b), only male mice were used in the following experiments. SQOR expression prevented the hypoxia-induced increase in the levels of sulfide and persulfide and the ratio of NADH/NAD$^+$ in the brain of male mice (Fig. 6f–h). Metabolomic analysis was used to investigate the effects of hypoxia on metabolites in the brain. Neuron-specific SQOR expression mitigated the hypoxia-induced increase in the ratios of lactate/pyruvate and succinate/fumarate in the brain of male mice that breathed 5.5% $O_2$ (Fig. 6I, j and Supplementary Fig. 8c, Supplementary Tables 4 and 5).

To further define the effects of enhanced sulfide oxidation on oxygen deprivation specifically in the brain, we examined the impact of SQOR expression on brain injury induced by global cerebral ischemia and reperfusion. Male mice were subjected to global cerebral ischemia induced by 20 min of bilateral carotid artery occlusion (2VO) followed by reperfusion. In control mice that were infected with AAV-GFP and subjected to 2VO and reperfusion, a number of Fluoro-Jade B (FJB)-positive dead neurons were observed in the hippocampal CA1 and CA3 regions (Fig. 6k–m) and cerebral cortex (Fig. 6n, o). In mice expressing SQOR in neurons, only a few FJB-positive neurons were found in the hippocampus and cortex after 2VO and reperfusion. The number of viable neurons counted in H&E-stained brain sections showed a reciprocal trend of the number of FJB-positive dead neurons in these brain regions (Supplementary Fig. 9). After 2VO and reperfusion, mice expressing SQOR exhibited better

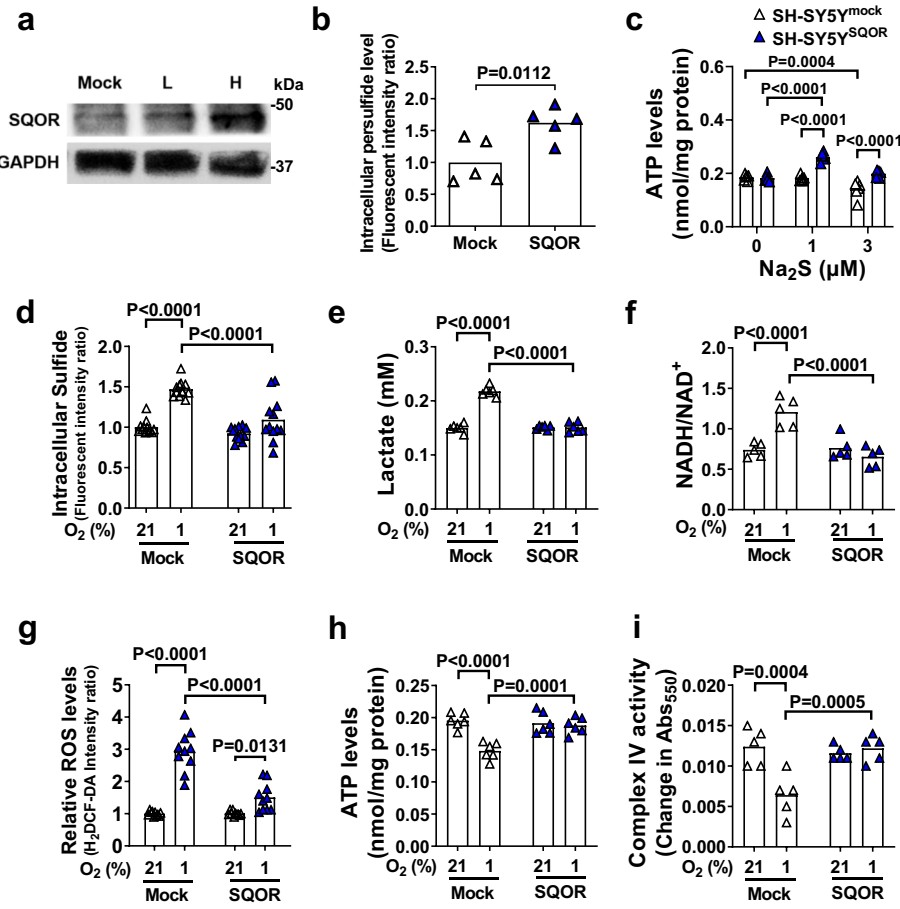

**Fig. 5 SQOR improves mitochondrial function in neuronal cells. a** Immunoblots of SQOR in SH-SY5Y with mock transfection or SQOR expression (L, H: low and high dose of transfection agent). Representative immunoblots of 2 independent biological replicates are shown. **b** Effect of SQOR expression on Intracellular persulfide level in SH-SY5Y cell at 21% $O_2$. $n = 5$ each. **c** ATP levels in mitochondria isolated from SH-SY5Y cells with or without SQOR expression treated with $Na_2S$ at 0, 1, or 3 μM in the medium ($n = 6$ each). **d** Intracellular $H_2S$ ($n = 12$ each), **e** lactate in cell culture medium ($n = 6$ each), **f** intracellular NADH/NAD$^+$ ratio ($n = 5$ each), **g** intracellular ROS ($n = 10$ each), **h** intracellular ATP ($n = 6$ each), and **i** complex IV activity ($n = 5$ each) in SH-SY5Y cells with or without SQOR expression in 21% or 1% $O_2$. Cells were exposed to hypoxia or normoxia for 3 h starting at 48 h after transfection. Data are presented as mean and individual values. A two-tailed unpaired $t$-test was performed for **b**. Two-way ANOVA followed by Sidak's correction for post-hoc comparisons were performed for **c**–**i**.

neurological function (Fig. 6p) and survival rate compared to control mice (survival rate at day 3, 9/9 vs 6/12, respectively, $P <$ 0.05). Taken together, these results suggest that neuron-specific SQOR expression prevents sulfide accumulation and impairment of mitochondrial respiration in the brain, improves survival in severe hypoxia, and prevents ischemic brain injury and death after global cerebral ischemia.

**Sulfide scavenging prevents ischemic brain injury**. The mechanism by which enhanced sulfide oxidation by SQOR increases ATP production is unknown. The predominant effect of sulfide oxidation may be to directly provide electrons to the ETC. Alternatively, the major effect of sulfide oxidation may be to prevent sulfide-induced inhibition of mitochondrial complex IV (Fig. 5i). Arguing against the first possibility, oxidation of sulfide as a direct source of electrons is energetically unfavorable, especially during oxygen shortage. Compared to oxidation of NADH or succinate (the canonical electron donors), sulfide oxidation requires three times more oxygen for the same electron transfer in the ETC[30]. We, therefore, hypothesized that the beneficial effects of SQOR-mediated sulfide catabolism during oxygen shortage are predominantly mediated by the avoidance of sulfide

accumulation, rather than the provision of electrons to ETC from oxidizing sulfide. To test this hypothesis, we examined the effects of pharmacological scavengers of sulfides on SH-SY5Y cells incubated in 21% or 1% $O_2$. Both the highly specific sulfide fluoroprobe/scavenger HSip-1 and the broad-spectrum scavenger hydroxocobalamin dose-dependently prevented the increase in the ratio of NADH/NAD$^+$ induced by 1% $O_2$ (Fig. 7a). Hydroxocobalamin attenuated the low oxygen-induced increase in SH-SY5Y intracellular sulfide levels (Fig. 7b), and improved survival of cells subjected to oxygen and glucose deprivation (Fig. 7c) in a dose-dependent manner. Although increased sulfide levels observed in primary cortical neurons of $Sqor^{\Delta N/\Delta N}$ mice (Fig. 3h) were associated with decreased oxygen consumption in isolated brain mitochondria obtained from $Sqor^{\Delta N/\Delta N}$ mice, scavenging sulfide by hydroxocobalamin partially restored OCR (Fig. 7d). These results suggest that scavenging sulfide can restore mitochondrial energy homeostasis even when sulfide oxidation is impaired.

In mice, global cerebral ischemia induced by 2VO decreased SQOR activity in the brain, leading to increased sulfide levels and the NADH/NAD$^+$ ratio (Fig. 7e–g). Administration of sulfide scavengers prevented sulfide accumulation and the increase in the NADH/NAD$^+$ ratio in the brain during global cerebral ischemia.

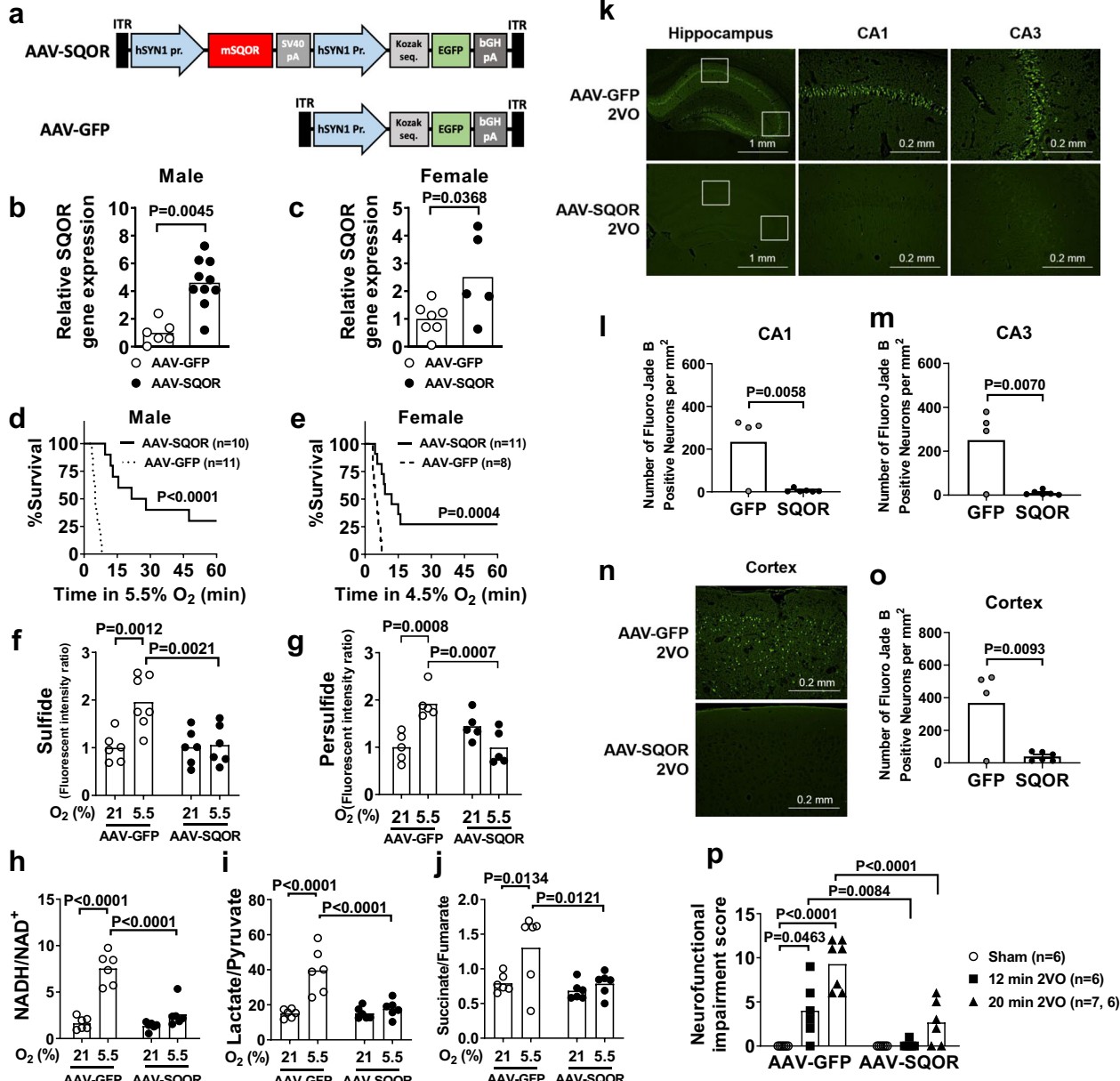

**Fig. 6 Effects of SQOR expression in the brain of mice. a** Structures of AAV containing mouse SQOR and enhanced GFP sequence under hSYN1 promoter (AAV-SQOR) and control AAV (AAV-GFP). SV40, simian virus 40. Relative SQOR mRNA expression levels in brains of 8-week-old **b** male ($n = 6, 10$) and **c** female ($n = 7, 5$) CD-1 mice transfected with AAV-GFP or AAV-SQOR on postnatal day 0. Survival curves of 8-week-old **d** male and **e** female CD-1 mice transfected with AAV-GFP or AAV-SQOR on postnatal day 0 and breathing 5.5% oxygen and 4.5% oxygen, respectively. Levels of **f** sulfide ($n = 6, 7, 6, 6$), **g** persulfide ($n = 5$ each), **h** the ratios of NADH/NAD$^+$ ($n = 6$ each), **i** lactate/pyruvate ($n = 6$ each), and **j** succinate/fumarate ($n = 6$ each) in male mice transfected with AAV-GFP or AAV-SQOR and breathed 21% or 5.5% oxygen for 3 min. **k** Representative photomicrographs of Fluoro-Jade B (FJB)-stained brain sections focusing on hippocampal CA1 and CA3 regions of male mice transfected with AAV-GFP ($n = 4$) or AAV-SQOR ($n = 6$) and subjected to 2VO and reperfusion. Number of dead neurons in **l** CA1 and **m** CA3 regions of mice transfected with AAV-GFP or AAV-SQOR and subjected to 2VO ($n = 4, 6$). **n** Representative photomicrographs of FJB-stained brain sections focusing on cerebral cortex region of male mice transfected with AAV-GFP ($n = 4$) or AAV-SQOR ($n = 6$) and subjected to 2VO and reperfusion. **o** Number of dead neurons in the cerebral cortex of mice transfected with AAV-GFP or AAV-SQOR and subjected to 2VO ($n = 4, 6$). **p** Neurofunctional impairment score of male CD-1 mice transfected with AAV-GFP or AAV-SQOR and subjected to sham surgery or 12 or 20 min of 2VO ($n = 6, 6, 7, 6, 7$). Data are presented as mean and individual values. A two-tailed unpaired $t$-test was performed for **b**, **c**, **l**, **m**, and **o**. Survival rates were estimated using the Kaplan–Meier method and a log-rank test was used to compare the survival curves between groups in **d** and **e**. Two-way ANOVA followed by Sidak's correction for post-hoc comparisons were performed for **f–j** and **p**.

Furthermore, treatment with hydroxocobalamin at 10 min after the onset of focal brain ischemia markedly attenuated ischemic brain injury and neurological dysfunction after permanent middle cerebral artery occlusion (MCAO) without reperfusion (Fig. 7h–j) and dose-dependently improved survival when mice were

exposed to 5.5% O$_2$ (Fig. 7k). In contrast, cyanocobalamin, a synthetic vitamin B12 that scavenges ROS, but not sulfide, did not improve the survival of hypoxic mice[44] (Fig. 7k). To further examine the ability of sulfide scavengers to protect against the effects of cerebral ischemia, mice were subjected to a more

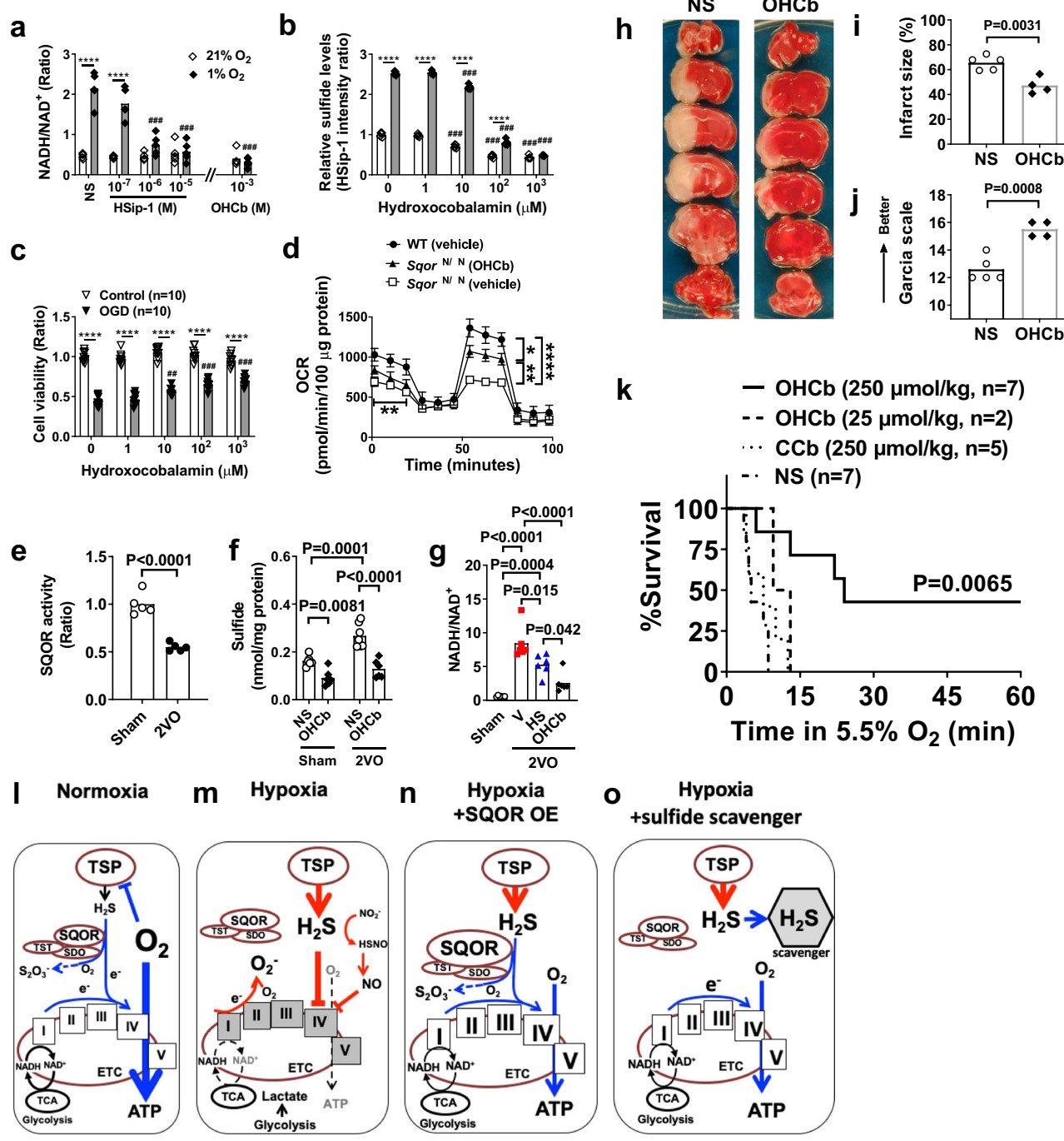

clinically relevant model of transient (60 min) MCAO and reperfusion and treated with the highly specific sulfide scavenger SS-20[45] 45 min after the onset of ischemia. Treatment with SS-20 during ischemia markedly decreased the infarct size and improved neurological function at 48 h after reperfusion without changing CBF (Supplementary Fig. 10). These results suggest that the prevention of sulfide accumulation is sufficient to maintain energy homeostasis and neuronal survival during acute oxygen deprivation in the brain.

## Discussion

In hypoxic tissues, not only is there a lack of oxygen, but there is also increased production of sulfide. In the brain, an organ with limited capacity to oxidize sulfide, hypoxia-induced sulfide

accumulation results in decreased oxidative phosphorylation and neuronal death (Fig. 7l, m). Our discovery that acceleration of sulfide oxidation in the murine brain induces tolerance to hypoxia prompted us to hypothesize that enhancement of sulfide catabolism may prevent sulfide accumulation and mitigate the impairment of mitochondrial energy homeostasis during oxygen shortage, resulting in decreased ischemic brain injury. In the current study, we observed that the degree of tolerance to hypoxia correlates with brain levels of SQOR in mice, rats, and 13LG squirrels. Decreasing the cellular level of SQOR or preventing the mitochondrial localization of the enzyme makes the brains more sensitive to oxygen shortage, whereas neuron-specific expression of SQOR makes mice highly resistant to hypoxia and cerebral ischemia (Fig. 7n). We also showed that scavenging sulfide maintains oxidative phosphorylation during oxygen shortage and

**Fig. 7 Effects of sulfide scavenger. a** NADH/NAD$^+$ ratio in SH-SY5Y cell lysates 3 h after incubation in hypoxia or normoxia with varying doses of HSip-1 or hydroxocobalamin (OHCb). $n = 5$ each. ***$P < 0.001$ vs normal saline (NS) at nomoxia. ###$P < 0.001$ vs NS at hypoxia. **b** Relative sulfide levels in SH-SY5Y incubated in normoxia or hypoxia in the presence of varying concentrations of OHCb. $n = 5$ each. **c** Cell viability of SH-SY5Y subjected to oxygen and glucose deprivation (OGD) in the presence of varying levels of OHCb, assessed by crystal violet assay. $n = 10$ each. ***$P < 0.001$, ##, ###$P < 0.1, 0.001$ vs OHCb 0 μM. **d** Oxygen consumption rate (OCR) was measured in isolated brain mitochondria from $Sqor^{\Delta N/\Delta N}$ mice with or without OHCb and compared to isolated mitochondria of wild-type littermates treated with vehicle ($n = 4$ each). **e** SQOR activity ($n = 5$ each) and **f** sulfide levels ($n = 6, 6, 7, 6$) in the brains of mice subjected to sham operation or 2VO and treated with saline or OHCb. **g** NADH/NAD$^+$ ratio in the brains of sham-operated mice or in mice 5 min after the start of 2VO treated with vehicle (V), HSip-1 (HS), or OHCb ($n = 6$ each). **h** Representative photographs of the TTC staining of coronal brain sections of male mice subjected to permanent MCAO and treated with normal saline (NS) or OHCb. **i** Brain infarct volume and **j** neurologic function in male mice after 2VO and reperfusion ($n = 5, 4$). **k** Survival curve of male CD-1 mice treated with OHCb, cyanocobalamin (CCb), or normal saline breathing 5.5% oxygen. Diagram illustrating working hypothesis on the role of SQOR and sulfide on mitochondrial energy production during **l** normoxia, **m** hypoxia, **n** hypoxia with SQOR expression, and **o** hypoxia with sulfide scavenger. TSP, transsulfuration pathway. TST, thiosulfate sulfurtransferase. ETHE1, ethylmalonic encephalopathy 1. TCA, tricarboxylic acid cycle, or Krebs cycle. Mitochondrial electron transport chain (ETC) complex are shown with roman numerals in boxes. Data are presented as mean and individual values. Two-way ANOVA followed by Tukey's or Sidak's correction for post-hoc comparisons were performed for **a**–**d** and **f**. Two-tailed unpaired $t$-test was performed for **e**, **i**, and **j**. One-way ANOVA followed by Tukey's correction for post-hoc comparisons were performed for **g**. Survival rates were estimated using the Kaplan–Meier method and a log-rank test was used to compare the survival curves between groups in **k**.

prevents ischemic brain injury in mice (Fig. 7o). Our observations illuminate a critical role of sulfide catabolism in facilitating the brain's resistance to acute oxygen deprivation. These results suggest that enhanced sulfide catabolism may be an effective therapeutic approach to the treatment of hypoxic or ischemic brain injury.

The role of sulfides in hypoxia or ischemia has been a focus of intense investigation in the last decade. Although sulfide levels increase during hypoxia and decrease after reoxygenation in several tissues, including the brain, the functional significance of the changes in sulfide levels in hypoxic tissue remained elusive[24,26,46,47]. While increasing sulfide levels by administration of exogenous sulfide donors prevented ischemic brain injury in some studies[48,49], decreasing sulfide levels by inhibition of sulfide-producing enzymes were also protective in other studies[50,51]. In the current investigation, we observed that breathing 5% oxygen for 3 min increased brain sulfide levels in mice to a similar extent as breathing H$_2$S at 80 ppm, which is sufficient to inhibit whole-body oxygen consumption (via inhibition of COX) by ~90%[31,32]. These results indicate that the levels of sulfide induced by a brief period of hypoxia are sufficient to inhibit COX in the brain. We also observed that sulfide pre-conditioning upregulates SQOR expression in the brain and abrogates the ability of breathing 5% O$_2$ or 80 ppm H$_2$S to increase sulfide levels in the brain. Sulfide pre-conditioning enhances the ability of brain mitochondria to withstand the inhibitory effects of sulfides on oxidative phosphorylation by upregulating SQOR and thereby makes mice tolerant to hypoxia. These results suggest that upregulation of SQOR confers beneficial effects during oxygen shortage in the brain.

Recent studies suggest that reactive persulfide has critical roles in mitochondrial bioenergetics[15]. In the current study, we observed that, along with sulfide, persulfide accumulated in hypoxic brains. Increased levels of SQOR prevented the hypoxia-induced persulfide accumulation in the brain, while persulfide accumulated in the brains and livers of hypoxic $Sqor^{\Delta N/\Delta N}$ mice. These observations indicate that, in addition to sulfide, persulfide is produced in hypoxic brains and catabolized by SQOR (either directly or indirectly after reduction to sulfide). Akaike and colleagues suggested that persulfide may serve as an electron acceptor from ETC[15]. Therefore, in hypoxic tissues with high SQOR levels, SQOR-dependent oxidation of sulfide and persulfide donate electrons to ETC, and persulfide may be reduced to sulfide by accepting electrons from ETC. SQOR may maintain mitochondrial energy homeostasis during hypoxia not only by preventing sulfide-induced COX inhibition but also by producing persulfide.

Because sulfide pre-conditioning and enhanced tolerance to hypoxia were associated with upregulation of SQOR in the brain, we posited that SQOR might also has a role in natural hypoxia tolerance. Previous studies of brain ischemia in a variety of animal models showed that, compared to male animals, females sustain less tissue damage than males after comparable insults[52,53]. The protective effect of the female gender is lost after ovariectomy but can be restored by estrogen supplementation[54]. In the current study, we found that the tolerance of female mice to hypoxia was associated with higher levels of SQOR in the brain compared to male mice. Isolated brain mitochondria from female mice were resistant to the inhibitory effects of sulfide on oxidative phosphorylation. Ovariectomy decreased brain SQOR levels and abolished hypoxia tolerance in female mice, whereas estrogen supplementation tended to increase brain SQOR levels and restored resistance to hypoxia in ovariectomized female mice. Silencing SQOR in the brain made female mice sensitive to hypoxia. These observations support the hypothesis that higher levels of SQOR in the brain enable hypoxia tolerance.

Several mammalian species exhibit robust resistance to severe oxygen deprivation at levels that are invariably lethal to other mammals. For example, 13LG squirrels can survive in 4.5% O$_2$, a level that is lethal to rats[40]. Nonetheless, the mechanisms responsible for the marked tolerance of ground squirrels to hypoxia were previously unknown. In the current study, we found that the brains of 13LG squirrels have a ~100-fold higher level of SQOR and a greater capacity to oxidize sulfide compared to mice and rats. Breathing 5% O$_2$ for 5 min increased levels of sulfide, lactate, succinate, fumarate, and the ratio of NADH/NAD$^+$ in the brains of rats, whereas hypoxia increased only succinate in the brains of 13LG squirrels. Succinate is the universal mitochondrial feature of tissue hypoxia[55]. The observed isolated increase of succinate confirms that the brains of 13LG squirrels are experiencing severe hypoxia, but protected from impairment of oxidative phosphorylation. Of note, knockdown of SQOR restored the ability of hypoxia to increase sulfide levels and the ratio of NADH/NAD$^+$ in the brains of 13LG squirrels. These results suggest that high SQOR levels in the brain of 13LG squirrels prevent hypoxia-induced sulfide accumulation and inhibition of oxidative phosphorylation during hypoxia.

Friederich and colleagues recently reported two families that have mutations in the $Sqor$ gene that severely decrease SQOR protein stability and enzyme activity[19]. Three individuals of these families developed Leigh syndrome-like encephalopathy triggered by acute infectious episodes and two of them died at young ages (younger than 8 years old). Liver and muscle tissue from one of

these patients showed markedly decreased complex IV activity. Although tissue sulfide levels were not measured in these individuals, these authors speculated that complex IV activity was inhibited by the accumulation of $H_2S$ induced by decreased SQOR activity. In the current study, $Sqor^{\Delta N/\Delta N}$ mice exhibited emaciation, ataxia, and short life span, recapitulating clinical features of Leigh syndrome. Primary cortical neurons of $Sqor^{\Delta N/\Delta N}$ mice showed increased $H_2S$ levels and decreased mitochondrial oxygen consumption under normoxic conditions. These observations suggest that SQOR is required for normal mitochondrial function and postnatal growth. In addition, $Sqor^{\Delta N/\Delta N}$ mice may represent a novel mouse model of Leigh syndrome.

The results in the study of $Sqor^{\Delta N/\Delta N}$ mice support the hypothesis that sulfide and persulfide may be used as electron donors in tissues with high native SQOR expression levels such as heart and liver. Exposure to 5.5% $O_2$ increased levels of sulfide and persulfide and the ratio of $NADH/NAD^+$ only in the brain, but not in the heart and liver, of wild-type mice. However, in $Sqor^{\Delta N/\Delta N}$ mice, breathing 5.5% $O_2$ increased levels of sulfide and persulfide and the ratio of $NADH/NAD^+$ in the heart and liver, and exaggerated the hypoxia-induced increase in sulfide levels and the ratio of $NADH/NAD^+$ in the brain. These observations suggest that basal levels of SQOR expression in the heart and liver, but not in the brain, are sufficient to catabolize sulfides and maintain oxidative phosphorylation when breathing 5.5% $O_2$ for 3 min. This hypothesis is corroborated by the lack of SQOR upregulation in the heart by sulfide pre-conditioning; basal SQOR levels in the heart are sufficient to catabolize sulfide during intermittent $H_2S$ inhalation. However, deficiency of SQOR in mitochondria in $Sqor^{\Delta N/\Delta N}$ mice increased the sensitivity of heart and liver to hypoxia. It is currently unknown how many electrons are donated to the ETC by oxidation of sulfides by SQOR, in addition to those donated by oxidation of canonical electron donors such as NADH and succinate. However, our results suggest that, compared to the brain, the ability of organs with high SQOR levels to use sulfides as electron donors makes these organs more resistant to oxygen deprivation.

Because sulfide oxidation to persulfide by SQOR donates electrons via $CoQ_{10}$ to complex III[21,22], disposal of excess sulfide and supplying electrons to ETC are inseparable functions of SQOR. Therefore, pharmacological sulfide scavenging may be only part of the explanation for the beneficial effects of SQOR-dependent sulfide oxidation in hypoxic brains. The partial restoration of oxygen consumption rates by sulfide scavenging in the brain mitochondria of $Sqor^{\Delta N/\Delta N}$ mice supports this hypothesis. Sulfide scavenging did not affect CBF during and after cerebral ischemia and sulfide pre-conditioning did not induce VEGF in the brain. These observations suggest that the beneficial effects of sulfide scavenging or SQOR during hypoxia are unlikely to be mediated by increasing CBF or promoting cerebral angiogenesis. It is of note that three sulfide scavengers (hydroxocobalamin, HSip-1, and SS-20) with distinct chemical structures exhibit robust protective effects in hypoxic or ischemic brains. Hydroxocobalamin (vitamin B12) is safe in humans and has been approved for the treatment of cyanide poisoning by the US FDA. Given the paucity of neuroprotective measures against hypoxic or ischemic brain injury, further studies examining the neuroprotective effects of sulfide scavengers are warranted.

There are limitations in the current study. We did not investigate how acute hypoxia increases sulfide levels in the brain. Sulfide levels are determined as a balance between sulfide synthesis and catabolism. It has been suggested that hypoxia decreases sulfide oxidation thereby increasing sulfide levels[24]. Our observation that 2VO decreased SQOR activity and increased sulfide levels in the mouse brain supports this hypothesis. While we used a sulfide donor $Na_2S$ as a "substitute" for hypoxia in

experiments in isolated mitochondria, response to sulfide and hypoxia may not overlap. Effects of SQOR in mitochondrial energy homeostasis remains to be determined under hypoxic condition. Although we did not find upregulation of canonical HIF-1α targets in the mouse brain when mice were exposed to hypoxia at 24 h after the last $H_2S$ inhalation, the role of HIF-1α in the induction of hypoxia tolerance of sulfide-pre-conditioned mice remains incompletely defined. $H_2S$ has been reported to inhibit[56] or activate HIF-1α[37]. It is possible that sulfide pre-conditioning induces SQOR via HIF-1α activation. Interaction between HIF-1α and sulfide oxidation pathway in hypoxia tolerance remains to be elucidated in future studies.

In conclusion, the current study demonstrates a critical role of sulfide catabolism in cellular energy homeostasis. Our results suggest that the relatively low level of SQOR makes most mammalian brains particularly vulnerable to oxygen deprivation, whereas higher levels of SQOR may contribute to the increased resistance to hypoxia exhibited by other tissues. Acceleration of sulfide catabolism or pharmacological sulfide scavenging during ischemia may make the brain more resistant to ischemia and provide a novel preventive or therapeutic approach to ischemic/hypoxic brain injury.

## Methods

Animal protocols were approved by the Massachusetts General Hospital Institutional Animal Care and Use Committee. Sprague-Dawley rats and CD-1 mice were purchased from Charles River Laboratory (Wilmington, MA). C57BL6/J mice were purchased from Jackson Laboratory (Bar Harbor, ME). $Sqor^{\Delta N/\Delta N}$ mice were generated in Tohoku University and transferred to and maintained at Massachusetts General Hospital. All mice were housed with a 12 h dark/light cycle at a temperature between 20 and 25 °C and a humidity between 40% and 60%. Thirteen-lined ground squirrels (13LGS) were purchased as weanlings from the University of Wisconsin Oshkosh Squirrel Colony. The design of experiments involving animal subjects followed the ARRIVE guidelines. To minimize variability, we used a randomized paired (a.k.a. matched pairs) design. We paired mice to two or more treatment groups on the basis of similar weight, age, delivery date, and when possible holding cage.

**Intermittent $H_2S$ inhalation**. Adult male C57BL6/J mice were used in these experiments. The mice assigned to sulfide pre-conditioning by $H_2S$ breathing were exposed to air containing 80 ppm of $H_2S$ (Airgas Inc., Radnor, PA) for 4 h on each of 5 consecutive days (Day 1 through Day 5), as previously described[33]. $H_2S$ concentration, as well as $FiO_2$, was continuously measured using a portable gas monitor (ITX Multi-Gas Monitor, Industrial Scientific Corporation, Oakdale, PA). Rectal temperature was measured pre- and post- $H_2S$ inhalation each day to examine the influence of $H_2S$ on body temperature.

**Measurements of $CO_2$ production**. Twenty-four hours after the last air or $H_2S$ inhalation, $CO_2$ production was measured in control or sulfide-pre-conditioned mice. After mice were placed in individual chambers (Buxco Electronics, Inc., Wilmington, NC), $CO_2$ production was measured with the LI-820 $CO_2$ Gas Analyzer (LI-COR Biosciences, Lincoln, NE). Mice breathed air for 1 h and exposed to $H_2S$ at 80 ppm or 5% oxygen[32].

**Hypoxic breathing and tissue harvesting in mice**. To measure hypoxia tolerance, 24 h after the last air or $H_2S$ inhalation, control or sulfide-pre-conditioned mice were placed into a clear 1 L plastic chamber. Thereafter, the chamber was flushed continuously at 1 L/min with gas mixtures to achieve the desired $FiO_2$. Using an oxygen monitor (Industrial Scientific Corporation, ITX multi-gas monitor equipped with sampling pump model ISP), we determined that chamber air turnover time was 228 ± 10 s. Mice were observed continuously for a maximum of 60 min during hypoxia exposures, and were removed from the chamber when mice exhibited signs of severe distress (wriggle or seizure, respiratory rates less than 6 per min, and incontinence) and euthanized with 5% isoflurane and counted as dead. We used different inspired oxygen levels in differing strains and gender of mice based on our pilot studies. For brain tissue harvesting, mice were anesthetized with isoflurane (induction 4%, maintenance 1.5%), intubated with a 20-G catheter, and mechanically ventilated (Harvard Apparatus, Mini Vent Ventilator Model 845) with a tidal volume of 8 μL/g body weight/stroke and 120 strokes/min. After 3 min of ventilation with air or hypoxic gas mixture, mice were euthanized with exsanguination and their forebrains were quickly harvested after perfusion with ice-cold PBS. Each hemisphere after removing cerebellum and olfactory bulbs were frozen in liquid nitrogen and stored at −80 °C until further assays.

**Generation of mitochondria-specific SQOR-deficient mice**. All experimental procedures conformed to the Regulations for Animal Experiments and Related Activities at Tohoku University, were reviewed by the Institutional Laboratory Animal Care and Use Committee of Tohoku University, and were approved by the President of the University. We generated *Sqor*-deficient mice as follows. We used B6D2F1 (C57BL/6NCr × DBA/2Cr F1) mice to obtain unfertilized eggs, and we fertilized these eggs in vitro. In vitro fertilization was performed according to a standard protocol with the B6D2F1 strain. After a 3-h culture of oocytes and sperm, the eggs were removed and cultured for 5 h until electroporation. The Genome Editor electroporator and LF501PT1-10 platinum plate electrode (length: 10 mm, width: 3 mm, height: 0.5 mm, gap: 1 mm) (BEX Co. Ltd., Tokyo, Japan) were used for electroporation. We introduced Cas9/gRNA ribonucleoproteins consisting of the Cas9 protein in complex with a targeting gRNA into fertilized eggs with electroporation, according to protocols previously reported[57], after which we transferred the eggs to oviducts of pseudo-pregnant females on the day of the vaginal plug. The sequence of the gRNA was designed as follows: 5′-TATCCTGGTGATGGCCCCAC-3′, located at exon 2 of the *Sqor* gene to generate *Sqor*-mutant mice. A founder mouse harboring the *Sqor*-mutant allele with a 14-bp deletion spanning the translation initiation codon in exon 2 (*Sqor*$^{\Delta N}$) was crossed with WT mice to obtain *Sqor* heterozygous (*Sqor*$^{\Delta N/+}$) mice. *Sqor*$^{\Delta N/+}$ mice were back-crossed to the C57BL/6j background for more than six generations. Genotyping was performed by means of PCR and gel electrophoresis of the PCR product. The genotyping primers were 5′-TGCTTCCTTTTAGCCTGATCTA-3′ and 5′-AAAACAGGCAAAGAGCCGGG CAC-3′.

**Establishment of immortalized MEFs**. WT and *Sqor*$^{\Delta N/\Delta N}$ immortalized MEFs (iMEFs) were established from mouse embryos at E13.5 and immortalized by the lentiviral introduction of SV40 large T antigen. WT and *Sqor*$^{\Delta N/\Delta N}$ embryos were obtained from *Sqor*$^{\Delta N/+}$ pregnant females mated with *Sqor*$^{\Delta N/+}$ males. Stable transformants were selected via 2 µg/mL puromycin, and three independent WT iMEF lines and *Sqor*$^{\Delta N/\Delta N}$ iMEF lines were established.

**Immunoblot analysis of MEFs**. Mitochondrial fractions were isolated from WT and mitochondria-specific *Sqor*-deficient iMEFs as previously described[58]. Briefly, cells were homogenized in isotonic buffer (10 mM HEPES, pH 7.4, 75 mM sucrose, 225 mM mannitol, and 2 mM EDTA) with a Teflon homogenizer for 50 strokes at 1600 rpm and centrifuged at $700 \times g$ at 4 °C for 10 min. The supernatants were centrifuged again at $5000 \times g$ at 4 °C for 10 min. The pellets were washed twice with the isotonic buffer and used as mitochondrial fractions. The soluble fractions obtained by this process were filtered through a 0.22-mm filter and used as cytosolic fractions. Western blot analysis was performed by using anti-SQOR (polyclonal antibody raised in house), anti-GAPDH (Santa Cruz Biotechnology, Santa Cruz, CA, USA; clone: sc-25778), and anti-SDHA (succinate dehydrogenase subunit A) (Abcam, Cambridge, UK; clone: ab14715). The polyclonal antibody for mouse SQOR was produced by immunizing rats with recombinant mouse SQOR. Samples were solubilized with Laemmli lysis buffer (125 mM Tris-HCl, 2% SDS, 20% glycerol, and 10% 2-mercaptoethanol, pH 6.8), loaded on SDS-PAGE, and transferred to polyvinylidene fluoride membranes (GE Healthcare, Little Chalfont, England). The membranes were blocked with Blocking One (Nacalai Tesque, Kyoto, Japan) at room temperature for 60 min and were incubated with primary antibodies (1/5000) diluted by Can Get Signal Immunoreaction Enhancer Solution 1 (TOYOBO, Osaka, Japan) at 4 °C overnight. After membranes were washed with TBST buffer (50 mM Tris-HCl, 150 mM NaCl, 0.1% Tween 20, pH 7.4), they were incubated with horseradish peroxidase-conjugated secondary antibody (1/5000) diluted by Can Get Signal Immunoreaction Enhancer Solution 2 (TOYOBO) at room temperature for 1 h. Immunoreactive bands were detected by using an ECL Prime Western Blotting Detection Reagent (GE Healthcare) with a luminescent image analyzer (ImageQuant LAS 500; GE Healthcare).

**The 13-lined ground squirrel hypoxia model**. We selected the 13LGS as a natural model of the hypoxia-tolerant animal as it amenable to laboratory housing, can be acquired from a USDA-certified breeding colony in the US, and is a rodent species that can be weight-matched for comparison with young Sprague-Dawley rats (a hypoxia intolerant rodent). However, this species of ground squirrel (*Ictidomys tridecemlineatus*) is also an obligate seasonal hibernator, with distinct phenotypes between summer and winter and requiring varied seasonal housing conditions. During the winter season, 13LGS will naturally fast and exhibit bouts of torpor over 2–3 week periods, in which body temperature declines to near ambient, and vital rates precipitously drop[59]. These bouts are punctuated by short periods of euthermy (12–18 h) in which vital rates and body temperature return to summer levels (a body temperature trace of a 13-lined ground squirrel hibernating in the laboratory is provided in Supplementary Fig. 11a). Ground squirrels were maintained in standard rodent housing (~22 °C) in summer (June–October) with a light cycle 14 L:10D and fed *ad libitum* on sunflower seeds, dehydrated vegetables, and dog food, then transferred to a temperature-controlled chamber and housed at 4 °C in the dark from October–March, as previously described[60]. To monitor the animals during the hibernation season and to identify interbout euthermic periods that occur between torpor bouts, core body temperature of the ground squirrels

was tracked using temperature telemeters (DSI TA-F10) implanted into the peritoneal cavity; this surgical procedure has been previously described[60].

The cerebral hypoxia tolerance of 13LGS has not yet been completely characterized, although several metrics indicate that ground squirrels demonstrate intrinsically superior tolerance than non-hibernators such as rats[61]. One important consideration for this model is the possibility that dramatic seasonal shifts in phenotype with the onset of hibernation may affect metabolic and biochemical parameters of interest, including cerebral hypoxia tolerance. As a result, we have primarily focused on summer tissue collections for 13LGS to avoid any confounding metabolic impacts of hibernation, although we have included some animals collected in winter to bolster sample sizes, which we note in Supplementary Table 6. Any tissues from winter animals included in this dataset were collected during the euthermic interbout arousal period (Supplementary Fig. 11a). We also conducted pilot studies to confirm that seasonal phenotypes did not affect key parameters of interest, such as the abundance and activity of the H$_2$S signal transduction pathway. 13LG squirrels did not show seasonal changes in responses to acute hypoxia breathing, in terms of the levels of sulfide, lactate, and the ratio of NADH/NAD$^+$ in the brain (Supplementary Fig. 11b–d). There were also no seasonal differences in the protein levels of enzymes that metabolize sulfide in the brains of 13LG squirrels (Supplementary Fig. 11e–k), nor differences in H$_2$S pathway enzyme activities.

**Hypoxic breathing and tissue harvesting in rats and 13LG squirrels**. To investigate cerebral hypoxia tolerance, forebrains of rats and 13LG squirrels were harvested during acute hypoxia exposures (FiO$_2$ 5% for 5 min). For tissue collections, rats and 13LG squirrels were anesthetized with isoflurane (induction with 5% isoflurane) with spontaneous breathing (maintenance with 1.5–3.5% isoflurane), then placed on a heating pad (FHC, Inc., DC Temperature Control System) to maintain the rectal temperature at 37.3 ± 0.3 °C which is the physiological temperature of healthy rats measured by a flexible wire probe (Physitemp Instruments, Inc., IT-18). We cannulated the femoral artery in a subset of rats and 13LGS (PE10 polystyrene catheter) to collect serial blood samples for PaO$_2$ analysis (LifeHealth, IRMA TRUPOINT Blood Analysis System). Inspired O$_2$ (FiO$_2$) was monitored throughout the procedure (Industrial Scientific Corporation, ITX multi-gas monitor equipped with a sampling pump model ISP). After breathing air (FiO$_2$ = 21%) or hypoxic gas (5% O$_2$ for 5 min), rats and 13LG squirrels were euthanized by exsanguination under deep anesthesia and perfused with ice-cold PBS via the left ventricle, then their forebrains were harvested. In normoxic animals, the cerebral cortex from the right hemisphere was dissected using forceps and processed immediately to measure mitochondrial oxygen consumption rate (OCR). In both normoxic and hypoxic animals, the left forebrain was snap-frozen after removing cerebellum and olfactory bulbs, then pulverized in liquid nitrogen to obtain a homogenized sample, and stored at −80 °C until biochemical assays.

To evaluate the effect of SQOR knockdown on cerebral outcomes of acute hypoxia exposure, we conducted a similar experiment in 13LG squirrels infected with AAV (AAV-SQOR$_{13LGS}$ or AAV-Ctrl; design of knockdown agent and AAV injection procedures are described below). These animals were induced with isoflurane in a chamber (5% isoflurane), then maintained on 1–3% isoflurane using mechanical ventilation (Harvard Apparatus, Inspira asv, 50 breaths/min, tidal volume 8.5–10 mL/kg, PEEP = 3 cm H$_2$O). Anesthesia with mechanical ventilation was supported by rocuronium bromide (1 mg/kg) and fentanyl (50 µg/kg) delivered IP. A femoral artery cannula was placed in all animals to monitor blood gases throughout the procedure. Rectal body temperature was maintained at 36.7 ± 0.3 °C which is the physiological temperature of healthy 13LGS. We also supported blood pressure during the procedure by delivering 2cc warmed saline SQ upon induction of anesthesia.

**Sulfide measurement by high-performance liquid chromatography (HPLC)**. To measure sulfide levels, we performed a fluorescent-based HPLC as reported previously[62]. Briefly, brain tissues were homogenized in 100 mM Tris-HCl buffer (pH 9.5, 0.1 mM diethylenetriaminepentaacetic acid) and centrifuged at $15,000 \times g$ × 10 min × 4 °C. Supernatant (30 µL) was added to the mixture of Tris-HCl buffer (70 µL) and monobromobimane (50 µL, 10 mM in acetonitrile), incubated at RT for 30 min in the dark, then 5-sulfosalicylic acid (50 µL, 200 mM) was added to stop the reaction and stabilize monobromobimane-labeled molecules. The solution was placed on ice for 10 min and centrifuged. The recovered supernatant was then analyzed using HPLC equipped with a multi λ fluorescence detector (Waters 2475, Waters, Inc.).

**Sulfide measurement by a H$_2$S-specific fluorescent probe**. To measure sulfide levels, we used H$_2$S-specific fluorescent probes HSip-1 and HSip-1 DA for tissue homogenates and cultured cells, respectively, as reported previously[63]. For tissue sulfide levels, brains were homogenized in HSip-1 solution (10 µM) in PBS that was deoxygenated by passing nitrogen bubbles. Lysates were centrifuged at $15,000 \times g$ for 10 min at 4 °C. Supernatant was transferred into a 96-well plate, incubated in the dark for 20 min at RT, then fluorescence was measured using a microplate reader (SpectraMax i3x, Molecular Devices, Inc.). For sulfide levels in cultured cells seeded in a 96-well plate, 10 µM HSip-1 DA was added to the culture medium immediately after exposure to 1% O$_2$ and incubated for 20 min at 37 °C in O$_2$

(5%)/air (95%). Cells were washed twice with warm Hank's Balanced Salt Solutions (HBSS) to remove excess probe in the medium and fluorescence was detected at $\lambda_{ex}/\lambda_{em} = 491/516$ nm.

**Measurement of tissue persulfide**. Tissue persulfide levels were estimated using a fluorescent probe, SSip1 DA. SSip1 DA detects sulfane sulfur which is a sulfur atom with six valence electrons and no charge ($S^0$) and a component of persulfide[64]. Briefly, tissues were harvested after perfusion with ice-cold PBS and frozen in liquid nitrogen instantly. Tissues were homogenized in PBS with SSip-1 DA at 5 μM and centrifuged after incubation at room temperature in dark for 20 min. Fluorescent of supernatant was measured using a microplate reader at $\lambda_{ex}/\lambda_{em} = 491$ nm/525 nm. Fluorescent intensity was normalized by tissue weight.

**Measurement of NADH/NAD⁺**. NADH/$NAD^+$ ratio in brain tissues or cultured cells was measured by using a NAD/NADH quantitation colorimetric kit (K337-100, BioVision, Inc.). Brain tissues were homogenized in extraction buffer and centrifuged. We measured NADH/$NAD^+$ ratio in the collected supernatant in accordance with the manufacturer's protocol. Cultured cells seeded in a 6-well plate were washed twice with warm HBSS, scraped with a cell lifter, transferred to a new tube, and homogenized. After centrifugation, the supernatant was used to measure NADH/$NAD^+$ ratio.

**Measurement of Lactate**. We measured lactate levels in brain tissue or cell culture medium using a colorimetric assay kit (Eton Biosciences, 1200012002). Briefly, ~25 mg of tissue was homogenized in 400 μL ethanol (Sigma-Aldrich, 187380) and centrifuged at $15,000 \times g$ for 10 min at 4 °C. We then measured lactate in the supernatant according to the manufacturer's protocol. Tissue lactate levels were normalized to protein content, measured by BCA assay. Lactate levels in the cell culture medium were assayed directly using the same assay kit.

**qPCR**. Total RNA was extracted from brain tissues of mice using TRIzol reagent (ThermoFisher Scientific). Briefly, a cerebral hemisphere was homogenized (Fisher Scientific, Power Gen 125) in 1 mL TRIzol. After adding 200 μL of chloroform, the sample was vortexed, incubated at RT for 3 min, then centrifuged at $12,000 \times g$ at 4 °C for 10 min. The top layer was carefully transferred to a new tube. After adding 500 μL isopropanol, RNA was collected by centrifugation and washed with 75% ethanol. cDNA was synthesized using MMLV-RT (Promega). SQOR, TST, ETHE1, SUOX, CBS, CSE, 3MST, VEGF, HO-1, EPO, GLUT-1, COX1, NDUFV1, and 18S ribosomal RNA transcript levels were measured by real-time PCR using a Realplex 2 system (Eppendorf North America). Primer sequences are listed in Supplementary Table 7. Changes in the relative gene expression normalized to levels of 18S rRNA were determined using the relative CT method.

**Immunoblots**. Protein levels in forebrain tissues of rats, 13LGS, and mice were determined by standard immunoblot techniques using primary antibodies against GAPDH (1:5000, Cell Signaling, 5174), vinculin (1:2000, Cell Signaling, 4650), SQOR (1:1000, Proteintech, 17256-1-AP), TST (1:1000, GeneTex, GTX114858), ETHE1 (1:1000, GeneTex, GTX115707), SUOX (1:2000, Abnova, H00006821-D01), CBS (1:5000, Cell Signaling, 14782), CSE (1:1000, Proteintech, 12217-1-AP), 3MST (1:1000, Sigma-Aldrich, HPA001240), VEGF (1:1000, Thermo Scientific, 710151), GLUT-1 (1:1000, Abcam, ab115730). Bound antibody was detected with a horseradish peroxidase-linked antibody directed against rabbit IgG (1:10,000, Cell Signaling, 7074), and was visualized using chemiluminescence with Lumigen ECL Ultra kit (Lumigen, TMA-6).

**Measurement of oxygen consumption rate**. Oxygen consumption rate (OCR) was measured using an Extracellular Flux Analyzer (Seahorse Biosciences, Seahorse XFp Analyzer). Mitochondria were isolated from brains of mice or cerebral cortex of rats and 13LGS as reported previously[65]. Mitochondrial isolation buffer (MSHE + BSA) was composed of 70 mM sucrose, 210 mM mannitol, 5 mM HEPES, 1 mM EGTA and 0.5% (w/v) fatty acid-free BSA (pH 7.2). Mitochondrial assay solution (MAS) was composed of 5 mM sodium pyruvate/sodium malate for mice (10 mM sodium succinate for rats and 13LGS instead), 70 mM sucrose, 220 mM mannitol, 5 mM KH2PO4, 5 mM MgCl2, 2 mM HEPES, 1 mM EGTA and 0.2% (w/v) fatty acid-free BSA, pH 7.2 at 37 °C. Cortex was minced with a razor blade, pestled 20 times with a plastic tissue grinder (Fisher Scientific, 14-222-358) in 600 μL of MSHE + BSA, then centrifuged at $800 \times g$ at 4 °C for 10 min to collect the supernatant. An aliquot of the supernatant was reserved for determination of protein concentration by BCA assay; this aliquot was centrifuged at $8000 \times g$ at 4 °C for 10 min and then the mitochondrial pellet was dissolved in MSA without BSA for analysis. The remaining supernatant was also centrifuged at $8000 \times g$ at 4 °C for 10 min but the mitochondrial precipitate was dissolved in MAS + BSA. Mitochondria were diluted in this solution, then seeded into a XFp cell culture miniplate (Seahorse Biosciences, Seahorse XFp FluxPak) at 2.5 or 5 μg/well. The plate was centrifuged at $2000 \times g$ at 4 °C for 15 min, then incubated 10 min at 37 °C. A seahorse cartridge with detection probes (Seahorse Biosciences, Seahorse XFp FluxPak) was loaded with adenosine 5′-diphosphate (ADP, final 5 mM for mice, 2.5 mM for rats, and 13LGS), oligomycine (final 2 μM for all animals), carbonyl

cyanide-p-trifluoromethoxyphenylhydrazone (FCCP, final 3 μM for mice, 4 μM for rats, and 13LGS), and antimycin A (final 4 μM for mice, 3 μM for rats, and 13LGS) in injection port A, B, C, and D, respectively. For rats and 13LGS, rotenone at 2 μM was added to wells before adding Na₂S. Mitochondrial OCR was measured immediately after Na₂S or vehicle (MAS + BSA) into wells containing mitochondria.

**Measurement of NADH, membrane potential, and oxygen consumption in mitochondria**. Briefly, mice were euthanized, reperfused with ice-cold PBS, and brains were harvested. Mitochondria were isolated from the brain as reported previously[66]. ADP- and Na₂S-induced changes in NADH, mitochondrial membrane potential, and oxygen consumption rate were measured using a custom-made spectrophotometer as previously described[38].

**Intracellular H₂S levels and cell viability**. Primary cortical neurons were prepared from the cortex of embryonic day E15 mice derived from $SQOR^{\Delta N/+}$ parents as described previously[67]. After seeding cells of each embryo into a 96-well plate, genotypes of the embryo were determined. Cells were used for experiments on 11 days after seeding. To measure intracellular H₂S levels, cells were incubated with a fluorescent probe HSip-1 DA (Dojindo Molecular Technologies, Inc.) at 5 μM in HBSS with 0.1% DMSO, washed with HBSS, subjected to OGD for 20 min, and analyzed by a microplate reader at the wavelength of $\lambda_{ex}/\lambda_{em} = 491/516$ nm as described previously[69]. Cell viability after OGD for 1.5 h followed by reoxygenation for 20 h was evaluated by crystal violet assay as reported previously[67].

**AAV-mediated gene transfer to the brain**. To investigate the role of SQOR in tolerance of female mice to hypoxia, we examined whether SQOR knockdown in the brain makes female mice hypoxia-sensitive. We designed a recombinant AAV9 vector encoding eGFP and an shRNA targeting mouse Sqor driven by the U6 promoter. AAV9-CMV-eGFP-U6-mSQOR-shRNA (AAV-shSQOR) was administered at $10^{10}$ viral particles per hemisphere via ICV injection to newborn female CD-1 mice on postnatal day 0 (P0), as described previously[39]. AAV9-CMV/CAG-eGFP (AAV-Ctrl) was injected into control female mice at the same viral particle number. Mice were used for the experiment 7–9 weeks after AAV injection.

To determine the impact of increased neuronal levels of SQOR on resistance to hypoxia and global cerebral I/R injury in vivo, we overexpressed SQOR in neurons using AAV-mediated gene transfer in mice. We administered AAV9-hSYN1-mSQOR-hSYN1-eGFP (AAV-SQOR) or AAV9-hSYN1-eGFP (AAV-GFP) at $10^{10}$ viral particles per hemisphere via intracerebroventricular (ICV) injection into the brain of newborn male CD-1 mice. We used the human synapsin 1 (hSYN1) promoter to drive neuron-specific expression of SQOR and GFP[39].

We confirmed successful gene transfer into the brain by immunofluorescence method against GFP using frozen coronal brain sections with the thickness of 12 μm as reported previously with some modification. We used chicken polyclonal anti-GFP antibody (AvesLabs, GFP-1010, 1:1000) and Alexa Fluor® 488 conjugated anti-chicken IgY antibody (Life Technologies, 1:300) as a primary and the second antibody[39]. Fluorescence images were obtained using epifluorescent microscopy with a Nikon Eclipse 80i microscope (Nikon Instruments).

**GC-MS-based measurements of brain metabolites**. Approximately 25 mg of brain tissues were homogenized in 300 μL of ice-cold methanol (Sigma-Aldrich, Fluka 14262) and 150 μL of H₂O in a round-bottom Eppendorf tube (Fisher Scientific, Power Gen 125). Next, 200 μL of ice-cold chloroform (Sigma-Aldrich, 650498) was added, the tube was vortexed, then the sample was shaken by a rotator (Labnet, Revolver™ Tube Mixer) at 4 °C for 20 min and centrifuged at $15,000 \times g$ for 10 min at 4 °C. The upper layer of the supernatant was transferred to a polypropylene tube. For selected experiments, the extraction water contained 2 mg/mL norvaline (Nor, Sigma-Aldrich, St. Louis, MO, USA) as an internal standard for normalization. For gas chromatography-mass spectrometry (GC-MS) analyses, dried metabolites were derivatized in 2% methoxyamine hydrochloride in pyridine (MOX, Sigma) followed by N-methyl-N-(tert-butyldimethylsilyl) tri-fluoroacetamide + 1% tert-butyldimethylchlorosilane (TBDMS, Sigma) according to published methods[68]. One microliter of the derivatized sample was injected into an Agilent 7890B GC with a 30 m DB-35MS capillary column in line with an Agilent 5977 T MS. A splitless inlet liner and inlet temperature of 270 °C were used with a helium carrier gas flow rate of 1 mL/min. Electron impact ionization energy was 70 eV, ion source temperature was 120 °C, and quadrupole temperature was 150 °C. Ion detection was performed in Scan Mode (m/z between 100 and 605). For analysis of tissue metabolites, ion counts were normalized to the total ion counts of the tissue sample.

**MS-based measurements of sulfide metabolites**
*Derivatization of samples by monobromobimane (MBB)*. Tissues were homogenized in Tris-HCl buffer (100 mM Tris-HCl, pH 9.5, 0.1 mM DTPA). Thirty μL of the sample was incubated with 100 μL of Tris-HCl reaction buffer (100 mM Tris, 0.1 mM DTPA, pH 9.5) and 50 μL of MBB (10 mM, dissolved in acetonitrile) under hypoxia (25 °C,1% O₂) for 30 min, and the reaction was stopped by addition of 50 μL sulfosalicylic acid (200 mM).

*RP-HPLC analysis of hydrogen sulfide.* Levels of hydrogen sulfide and thiosulfate in brain tissues were measured by RP-HPLC as stable products sulfide-dibimane (SDB) and thiosulfate bimane (TSB), respectively, as previously described[70]. The separation of SDB and TSB was performed using Shimadzu Prominence 20 A equipment with RF-10AXL ($\lambda_{ex}$ 390 mm and $\lambda_{em}$ 475 mm) and an Eclipse XDB-C18 column (4.6 × 250 mm, 5 μm) with gradient elution by 0.1% (v/v) tri-fluoroacetic acid in acetonitrile. Typical retention times of SDB and TSB were around 16.5 and 9.5 min, respectively. Free sulfide and thiosulfate levels were calculated according to their standard curves.

Cysteine, homocysteine, and glutathione levels were measured by multiple reaction monitoring (MRM) using Acquity UPLC system coupled to XEVO TQ (Thermo Scientific) with electrospray ionization (ESI(+)). UPLC separation was performed on a BEH C18 column (2.1 mm × 100 mm, 1.7 μm particle size) (Waters, Mississauga, ON, Canada). Mobile phases for LC were water (0.1% TFA, A) and acetonitrile (0.1% TFA, B), and the flow rate was set to 0.3 mL/min. Gradient was as follows: 0 min, 15% B; 0–10 min, 45% B (linear); 10–11 min, 95% B (linear) and hold for 1.0 min; 12–13 min, 15% B (linear) and hold for 2.0 min. Data were collected in MRM mode by screening parent and daughter ions simultaneously, cone voltage was set depending upon each specific MRM for each metabolite, and the dwell time was automatically set by the MassLynxTM 4.1 software. The optimum collision energy and precursor/production ions for thiol derivatives were summarized in Supplementary Table 8. Fold changes of cysteine, homoCys, and glutathione levels were performed using the peak area ratios of signature productions to corresponding stable $S^{34}DB$ (internal standards).

**Enzyme activities of SQOR.** SQOR activity in the brain was measured as reported previously with some modification[71]. Approximately 50 mg of forebrain from rats, 13LGS, or a whole forebrain hemisphere of mice were homogenized in 400 μL in RIPA lysis buffer (Boston BioProducts, BP-115). After centrifugation, the protein concentration of the supernatant was adjusted to 10 mg/mL with RIPA buffer, then adjusted to 0.5 mg/mL with PBS (pH 7.4) containing 22.2 μM coenzyme Q1 (Sigma-Aldrich, C9538). Na$_2$S was quickly dissolved (final concentration 800 μM) in 0.2 M Tris-HCl buffer (2.4 mM sodium sulfite, 0.1 mM DTPA, pH 9.5) deoxygenated with nitrogen bubble. Diluted supernatant (900 μL) was combined with 100 μL Na$_2$S solution in a trUView Cuvette (Bio-Rad, 170-2511), immediately prior to the start of the kinetic assay. We calculated SQOR activity as absorbance change over 30 s at 275 nm (Bio-Rad, SmartSpec Plus).

**Design of AAV-shRNA for SQOR knockdown in 13LG squirrels.** To examine the role of SQOR (SQOR) in 13LG squirrels, we performed brain SQOR knockdown employing the shRNA technique combined with AAV infection. First, we validated a species-specific target sequence for SQOR knockdown using primary myoblasts isolated from 13LG squirrels. Primary myoblasts of 13LG squirrels were isolated and stored in a cryogenic tank in the same manner as reported in mice previously[72]. Primary myoblasts were seeded in collagen-coated 6-well plates and transfected with shRNA when cells reached 40–60% confluence in F-10 medium supplemented with 20% FBS and penicillin/streptomycin. We designed and synthesized four shRNA plasmids (A, B, C, and D) with target sequences shown in Supplementary Table 3 using a BLOCK-iT™ U6 RNAi Entry Vector Kit (Life Technologies, catalog # K494500) and examined their knockdown efficiency. shRNA transfection was performed using OptiMEM (Life Technologies, catalog # 31985-070) and lipofectamine 2000 (Life Technologies, catalog # 11668019) according to the manufacturer's protocol and cells were harvested at 48 h after transfection. SQOR protein levels after transfection were determined using immunoblotting (Supplementary Fig. 7e). We synthesized an AAV for in vivo SQOR knockdown (AAV-shSQOR$_{13LGS}$) based on the sequence of shRNA C, which provided the highest degree of ex vivo knockdown in myoblasts. To determine the most effective AAV serotype for use in the 13LGS brain, we compared the gene transfer efficiency for four AAV serotypes (AAV2, 4, 8, and 9) using an Adeno-associated virus Serotype Testing Kit with eGFP expression as a tag protein (GeneCopoeia, catalog # AA320) to determine the most effective AAV to transfer genes into brains of 13LGS. Brains of 13LGS were harvested one week after AAV injection ICV (procedure is described below) and analyzed for eGFP expression by qPCR with a primer set (GeneCopoeia, catalog # SCQP00002). Based on the brain expression levels of eGFP, we chose AAV9 as a carrier virus of shRNA for SQOR knockdown (Supplementary Fig. 7f).

**Brain SQOR knockdown in 13LG squirrels.** Weanling, non-reproductive 13LG squirrels (body weight: 192–312 g, both sexes) were anesthetized with isoflurane in an induction chamber (5% isoflurane) and fixed on a stereotaxic frame with ear bars (ASI instruments, catalog # MM-8000/3). Anesthesia was maintained with 1–3% isoflurane using an anesthesia mask for rats (ASI instruments, catalog # RA-100G). Fur was removed from the top of the head and the skin was prepped for incision with standard aseptic technique. After application of local analgesia to the skin site, the skull was exposed via a 2 cm incision, then bilateral burr holes were made in the skull at the coordinate of A/P = +7.0 mm from the interaural line, M/L = ±2.0 mm using a micro-drill (Braintree Scientific, catalog # MD-1200). Coordinates were based on the 13LG squirrel brain atlas[73]. The needle of a microsyringe (Hamilton, catalog # 80530) was inserted to a depth of D/V = +13.5

mm from the interaural line and, then, AAV-shSQOR$_{13LGS}$ or AAV-Ctrl (2.5 × 10$^{11}$ GC/hemisphere) in 12.5 μL of saline was slowly injected. After bilateral injection and removal of the needle, holes of the skull were covered with a bone wax and the skin incision was sutured closed. Animals were returned to their home cage and used for hypoxia experiments two weeks later.

**Bilateral common carotid artery occlusion (2 vessel occlusion, 2VO).** To investigate the role of SQOR in cerebral ischemia and reperfusion injury, CD-1 mice with or without AAV9-mediated SQOR expression were subjected to global cerebral ischemia and reperfusion injury induced by 2VO in accordance with a previous report with some modifications[74]. Briefly, mice (male, 7–8 weeks old) were anesthetized by ketamine (80 mg/kg)/xylazine (12 mg/kg) IP, endotracheally intubated with a 20-G i.v. catheter, and breathed air during the procedure. Bilateral common carotid arteries were occluded with microclips (Fine Science Tools Inc., 18055-04) for 12 or 20 min and reperfused by removing microclips. Rectal temperature was monitored using a thermistor with a flexible wire (Physitemp Instruments, Inc., IT-18) and maintained at 37.5 °C using a heating pad (FHC, Inc., DC Temperature Control System) until 15 min after the start of reperfusion. Sham-operated mice were subjected to the same procedures except for occlusion of common carotid arteries. Mice were given 1 mL of lactated Ringer solution containing 5% dextrose IP after surgery daily. Neurofunctional impairment was assessed and scored as previously reported[75]. Survival rate for 3 days after surgery of control mice and mice overexpressing SQOR were 50% (6/12) and 100% (9/9), respectively. The investigator (EM) who performed 2VO operation and neurofunctional impairment assessment was blinded to the identity of the mice.

**Histological analysis after 2VO.** Mice were anesthetized with pentobarbital (200 mg/kg, IP), perfused with 4% paraformaldehyde (PFA) in ice-cold phosphate-buffered saline (PBS), and the whole brain was harvested 3 days following 20 min 2VO. Brains were further fixed in 4% PFA in 15 mL conical tubes until brains settle on the bottom. After dehydration with ethanol, 8 μm coronal brain sections were cut with a rotary microtome (Leica Biosystems, Inc.). Sections were stained with hematoxylin and eosin and microphotographs were taken using a microscope. The number of viable neurons in hippocampal CA1/CA3 regions and cerebral cortex were manually counted. For each brain region, 4–6 randomly selected levels were analyzed by an investigator who was blinded to treatment.

**Fluoro-Jade B staining after 2VO.** To evaluate neuronal degeneration in cerebral cortex and hippocampus at 72 h after 20 min of 2VO in male mice transfected with AAV-GFP or AAV-SQOR, paraffin-embedded coronal brain sections were stained with Fluoro-Jade B according to the previously described procedure[76]. Fluoro-Jade B is a fluorochrome used for labeling degenerating neurons. Paraffin sections were cut at a thickness of 8 μm. Six randomly selected brain sections per mouse were stained with Fluoro-Jade B. Briefly, deparaffinized, and rehydrated brain sections were incubated in 0.06% potassium permanganate solution for 10 min. After rinsing, sections were incubated in 0.0004% of Fluoro-Jade B staining solution for 20 min. Following Fluoro-Jade B staining, slides were rinsed, air dried, and coverslipped. Thereafter, 3 brain regions (cerebral cortex, CA1, and CA3) in each brain section were analyzed by epifluorescent microscopy with a Nikon Eclipse 80i microscope (Nikon Instruments). Fluorescein isothiocyanate (FITC) filter system was used to visualize Fluoro-Jade B labeling. One image per region per brain section was captured, and thus total of 6 images per region per mouse was analyzed. The number of Fluoro-Jade B positive cells per 1 mm$^2$ in each region was calculated with ImageJ 1.49v (NIH) using the plug-in Cell Counter in a blinded manner.

**Cell culture and transfection studies in SH-SY5Y.** We cultured human neuroblastoma cell line SH-SY5Y in DMEM/F12 50/50 Mix (Cellgro by Mediatech, Inc.) supplemented with 10% fetal bovine serum (FBS), 100 IU/mL penicillin, and 100 μg/mL streptomycin at 37 °C in air with 5% CO$_2$. For SQOR expression, SH-SH5Y cells were seeded in 96-well, 6-well or 10 cm diameter plates, and cultured for 24–48 h until 30–50% confluent. We transfected the cells with plasmid DNA for mock (Invitrogen, 12536017) or SQOR expression (Harvard Medical School, HsCD00043463) at 0.2, 4, or 16 μg/well using OptiMEM (Invitrogen, 31985-062) containing Lipofectamine 2000 reagent (Invitrogen, 11668) at 0.2, 4, or 16 μL/well. Cells were used for experiments 48 h after transfection.

For SQOR silencing, SH-SY5Y cells were seeded in 6-well plates, transfected with 20 nM control or SQOR knockdown siRNA (Dharmacon Inc., D-001810-10-20 or L-008271-01-0010, respectively) using OptiMEM with Lipofectamine RNAiMAX reagent (Invitrogen, 13778075), and used for assays after 48 h. We included a glycolysis inhibitor (10 mM 2-deoxy-D-glucose) in the cell culture medium during hypoxia in some assays to attenuate the influence of high glycolytic activity in these cells.

**Intracellular persulfide measurement.** Intracellular persulfide levels in SH-SY5Y cells were measured using a sulfane sulfur-specific probe, SSP4 (provided by Dr. Ming Xian, also available at Dojindo Molecular Technologies, Inc.) as reported previously[77]. Briefly, after cells get 30–50% confluent in a 96-well plate, cells were subjected to transfection with plasmids for SQOR expression or mock transfection.

Cells were incubated with SSP4 at 10 µM (0.1%DMSO) at 37 °C for 20 min, washed with HBSS and measured fluorescent intensity at $\lambda_{ex} = 482$ nm, $\lambda_{em} = 515$ nm using a microplate reader.

**Hypoxia and oxygen/glucose deprivation.** For in vitro hypoxia experiments, SH-SY5Y cells were incubated for 3 h in DMEM 37 °C either in air (control) or 1% $O_2$ (hypoxia) in an air-tight chamber (Billups-Rothenberg, Modular Incubation Chamber), both with 5% $CO_2$. For oxygen-glucose deprivation (OGD), SH-SY5Y cells were incubated for 3 h at 37 °C in RPMI1640 (Cellgro, 10-043-CV) in the air or 1% $O_2$, both with 5% $CO_2$.

**Relative mtDNA level.** The mtDNA copy number on the brain and heart was determined by qPCR analysis. Amplification curves for the COX I gene of the mtDNA and the NDUFV1 nuclear DNA gene were obtained to measure the relative mtDNA/nuclear DNA ratio. The reaction was initiated at 94 °C for 10 min, followed by 40 cycles through 94 °C × 10 s, 60 °C × 30 s, and 94 °C × 10 s. All reactions were run in duplicate[78,79].

**Hemoglobin and hematocrit.** Immediately after the mice were anesthetized, their blood will be aspirated from the heart with a heparin syringe, hemoglobin, and hematocrit were measured using a blood gas analyzer (ABL 800 FLEX; Radiometer Medical, Bronshoj, Denmark).

**Measurement of oxygen dissociation curve (ODC).** Oxygen dissociation curves were determined with a Hemox-Analyzer (TCS Scientific Corp., New Hope, PA)[80]. Collected whole blood (20 µL) was diluted with HEMOX solution (3 mL, TCS Scientific Corporation, New Hope, PA) and the anti-foaming agent (6 µL, TCS Scientific Corporation). HEMOX solution contains N-[Tris(hydroxymethyl) methyl]-2-aminoethanesulfonic acid (TES, 30 mM), sodium chloride (135 mM), and potassium chloride (5 mM) in water (pH 7.4). The ODC of the diluted blood sample was measured using a HEMOX analyzer (TCS Scientific Corporation) with the sample maintained at 37 °C. The partial pressure of oxygen at which 50% of Hb is oxygenated was determined as $P_{50}$ from ODC.

**Ovariectomy and 17β-estradiol replacement.** WT CD-1 female mice (10-14 weeks old) were subjected to ovariectomy or sham operation as reported previously.[81] A silicon capsule filled with sesame oil with or without 17β-estradiol (Sigma-Aldrich, E8875) at 36 µg/mL was implanted subcutaneously on the back of mice. Mice were used for experiments after 2 weeks.

**Enzyme activities of sulfide-producing enzymes.** Enzyme activity of cystathionine beta synthase (CBS), cystathionine gamma lyase (CSE), and 3-mercaptopyruvate sulfurtransferase (3MST) in the brains of rats and 13LGS were measured by monitoring $H_2S$ production from brain lysates in the presence and absence of chemical inhibitor of each enzyme. Approximately 25 mg of the brain was homogenized in 400 µL of 30 mM HEPES buffer (1 mM EDTA, pH 7.4) and centrifuged at $15,000 \times g$ at 4 °C for 10 min. Supernatant was transferred to a new tube and diluted with HEPES buffer to adjust protein concentration to 1 mg/mL. The reaction cocktails were based in HEPES buffer and contained substrates and cofactors specific to each enzyme reaction and a $H_2S$-specific fluorescent probe (20 µM HSip-1) were prepared with or without an inhibitor for each enzyme. The reaction cocktails included: for CBS, 20 mM homocysteine, cysteine 20 mM, 0.1 mM pyridoxal-phosphate, 0.2 mM S-adenosyl-L-methionine; for CSE: 20 mM cysteine, 0.1 mM pyridoxal-phosphate; and for 3MST: 0.2 mM 3-mercaptopyruvate, 0.2 mM dithiolethione. Inhibitors were: 2 mM aminooxyacetic acid (AOAA) for CBS; 2 mM D,L-propargylglycine (PAG) for CSE; and 0.2 mM compound 3 for 3MST[82]. Immediately after mixing 50 µL of HEPES reaction buffer and 50 µL of supernatant in a 96-well plate, we monitored fluorescence at $\lambda_{ex}/\lambda_{em} = 491/516$ nm over 30 min, then calculated relative enzyme activity using fluorescence of the reaction with inhibitor minus fluorescence without inhibitor for each enzyme.

**Measurements of ATP.** ATP levels in mitochondria isolated from SH-SY5Y cells were measured using a Luminescent ATP Detection Assay Kit, using the manufacturer's protocol (Abcam, ab113849). Mitochondria were isolated using a mitochondria isolation kit (ThermoFisher Scientific, 89874), then seeded in a 96-well plate at 2 µg/well. Mitochondria were then incubated with 1 µM sodium sulfide ($Na_2S$, Sigma-Aldrich, 208043) or vehicle (PBS) at 37 °C for 5 min in the presence of 10 mM sodium succinate, 5 mM adenosine 5′-diphosphate and 4 µM rotenone in MAS + BSA buffer (described in a method section for the Seahorse apparatus), and lysed by adding detergent. For hypoxia exposure, ATP was measured in SH-SY5Y cells transfected with plasmids for 48 h, then exposed to 21 or 1% $O_2$ for 3 h in the presence of 10 mM 2-deoxyglucose in DMEM.

**Measurements of complex IV activity.** Complex IV activity of SH-SY5Y cells were measured using a Complex IV Human Enzyme Activity Microplate Assay Kit (Abcam, ab109909). We measured complex IV activity in SH-SY5Y cells seeded into a 10 cm diameter dish, transfected with plasmids for mock or SQOR expression

over 48 h, then exposed to 1% or 21% $O_2$ for 3 h. Procedures were in accordance with the manufacturer's protocol, and in a dark room to prevent signal loss.

**Permanent middle cerebral artery occlusion (pMCAO).** WT CD-1 mice (male, 8–10 weeks old) were subjected to permanent MCAO as reported previously with some modifications[83]. Briefly, mice were anesthetized with 1.5% isoflurane in oxygen. Mice breathed 100% oxygen with spontaneous breathing from the induction of anesthesia to 15 min after MCA reperfusion. A 7-0 monofilament (Doccol Corp., catalog # 702156PK5Re) was inserted via an internal carotid artery to occlude the middle cerebral artery (MCA). Saline or OHCb at 250 µmol/kg was administered IV at 10 min before the onset of MCAO. Rectal temperature was monitored and kept at 37 ± 0.5 °C using a heating pad under mice during surgery until mice recover from anesthesia. Changes in cerebral blood flow (CBF) were monitored during surgery to confirm the occlusion of MCA using a laser Doppler flowmetry (Moor Instruments, Inc.) probe positioned at 2 mm posterior, 5 mm lateral from the Bregma. All mice used in this study exhibited the reduction of CBF over 80% from the baseline after MCA occlusion. Taking circadian rhythm into account, all mice were subjected to surgery in the daytime (9:00 to 17:00). Mice received 1 mL of 5% dextrose-enriched lactated Ringer's solution (ip) daily to avoid dehydration. After evaluating neurological function according to the previously reported criteria[84], brains were harvested to measure cerebral infarct volume using triphenyltetrazolium chloride (TTC) staining at 24 h after onset of MCAO.

**Transient middle cerebral artery occlusion (tMCAO).** WT CD-1 mice (male, 8–10 weeks old) were subjected to transient MCAO for 60 min and reperfusion for 48 h according to a previous report with some modification[85]. Briefly, mice were anesthetized with 1.5% isoflurane in oxygen. Mice breathed 100% oxygen with spontaneous breathing from the induction of anesthesia to 15 min after MCA reperfusion. Rectal temperature was monitored and kept at 37 ± 0.5 °C using a heating pad under mice during surgery until mice recover from anesthesia. Changes in cerebral blood flow (CBF) were monitored during surgery to confirm the occlusion of MCA using a laser Doppler flowmetry (Moor Instruments, Inc.) probe positioned at 2 mm posterior, 5 mm lateral from the Bregma. Common carotid artery (CCA) was occluded with a microclip and a 7-0 monofilament was inserted via the isolated external carotid artery (ECA) through an internal carotid artery to occlude MCA. MCA was reperfused by removing a monofilament at 60 min after starting MCAO. CCA was reperfused after ECA closure by removing a microclip at 15 min after MCA reperfusion. Saline or SS20 at 250 µmol/kg was administered IV at 15 min before MCA reperfusion. Taking circadian rhythm into account, all mice were subjected to surgery in the daytime (9:00 to 17:00). Mice received 1 mL of 5% dextrose-enriched lactated Ringer's solution (ip) daily to avoid dehydration. After evaluating neurological functional score[84], brains were harvested to measure cerebral infarct volume using TTC staining at 48 h after MCAO.

**Statistics.** Data for continuous variables are presented as means with plots of all individual values. An unpaired two-tailed Student's t-test or Mann–Whitney U test was used to compare two independent groups, as appropriate, for continuous variables. One-way or two-way analysis of variance (ANOVA) followed by Sidak's correction for post-hoc comparisons and Kruskal–Wallis test followed by Dunn's multiple comparisons were used for post-hoc comparisons for normally distributed data and non-normally distributed data, respectively. In three group comparisons, post-hoc comparisons were performed only in comparisons between two groups of each set. Survival rates were estimated using the Kaplan–Meier method and a log-rank test were used to compare the survival curves between groups. Significance was considered at the level of $P < 0.05$. GraphPad Prism 8.4.3 (GraphPad Software Inc., La Jolla, CA, USA) was used for statistical analyses.

**Reporting summary.** Further information on research design is available in the Nature Research Reporting Summary linked to this article.

## Data availability

The authors declare that the data supporting the findings of this study are available within the paper and its Supplementary Information files. Any remaining data that support the results of the study will be available from the corresponding author upon reasonable request. Source data are provided with this paper.

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

## Acknowledgements

This work was supported in part by Grants-in-Aid for Scientific Research (S and C) from the Ministry of Education, Sciences, Sports, and Technology (MEXT), Japan, to M.M. (grant no. 19K07341), T.I. (20K07306), T.M. (19K07554), and T.A. (18H05277). M.N. and T.A. were supported by CREST, Japan Science and Technology Agency (JST) (grant no. JPMJCR2024). H.M. was supported by Japan Agency for Medical Research and Development (AMED) (grant no. JP21gm5010002). D.N.A. was supported by the National Institute of Neurological Disorders and Stroke (NINDS) (grant no. R01NS096237). C.G.K. was supported by the National Heart, Lung, and Blood Institute (NHLBI) (grant nos. R01HL113303 and R01HL149264) and the National Institute of General Medical Science (NIGMS) (grant no. P20GM121307). F.I. was supported by the NHLBI (grant no. R01HL101930) and the NINDS (grant no. R01NS112373). F.I. and M.X. were supported by the NINDS (grant no. R21NS116671). A.H. and F.I. were supported by the National Science Foundation (grant no. #1929592). We thank Dr. Sandra L. Martin for contribution of 13-lined ground squirrel tissues and Drs. S. Martin and Katharine R. Grabek for the identification of brain-specific *Sqor* transcript isoform sequences for the 13-lined ground squirrel. Authors thank Drs. Vamsi Mootha and Warren M. Zapol for valuable discussion and support.

## Author contributions

E.M., S.H., A.G.H., and F.I. conceived of this study. E.M., M.M., S.H., S.K., R.M.H.G., Y.M., F.N., L.T., A.M., T.I. (Ida), T.M., D.R.F., B.C., N.M., Y.Y., A.B., R.L., T.T., T.I. (Ikeda), A.N., H.I., X.S., and A.G.H. performed experiments and data analysis. K.H., M.X., T.A., and H.M. provided resources. D.N.A., B.A.O., M.N., C.G.K., T.A., A.G.H., H.M., and FI. supervised the study. D.B.B. edited the manuscript. E.M., T.A., A.G.H., H.M., and F.I. wrote the manuscript with consultation from all authors.

## Competing interests

The authors declare no competing interests.
