## [Peer Review File · Nature Communications]

REVIEWER COMMENTS

Reviewer #1 (Remarks to the Author):

Hydrogen sulfide is a member of a small family of labile biological signaling gasotransmitters. H₂S plays a broad range of physiological and pathophysiological functions, including induction of angiogenesis, regulation of neuronal activity, vascular relaxation, glucose homeostatic regulation, and protection against I/R injury in the heart, liver, kidney, lung, and brain.

In this MS, authors serendipitously found intermittent breathing in H₂S for 5 days pre-conditioned mice against cerebral ischemic insult. SPC mice will not show hypo-metabolic effects than the acute and chronic inhalation of H₂S (control). Chronic exposure to H₂S is lethal. Female mice showed a higher degree of resistance to hypoxia since it has a higher level of SQOR in the brain, while mice lacking the expression of mitochondrial SQOR showed higher sensitivity to hypoxia. Comparisons between hypoxia-tolerant ground squirrels and rats or mice were also made to highlight the higher capacity to catabolize sulfide in the brain of ground squirrels since they express higher levels of SQOR compared to mice and rats. SSQOR knocking down using AAV shRNA reduced ground squirrels' resistance to hypoxia. The authors went on to show SQOR is responsible for ischemic brain injury in two cerebral ischemia models: 2VO (BCAO) and permanent pMCAO.

These findings are interesting, and some are surprising. Intermittent inhalation of H₂S is protective with a pre-conditioning effect is novel, so is the deletion of SQOR studies. The MS is written in a logical manner. The strength of the manuscript is that multiple approaches were used to determine the role of SQOR in H₂S catabolism and its relevance to hypoxia tolerance in several cerebral ischemia models. However, parts of the results are just too brief, and not enough details to support the statement made.

Major concerns are:

1. The lack of investigation on the potential mechanisms between chronic exposure to H₂S and intermittent chronic exposure to H₂S. What if a chronic, but non-lethal low dose treatment with H₂S was used. Does that provide pre-conditioning towards hypoxia tolerance? Increased tolerance to H₂S cannot only be assessed using lethality as a sole or major criterion. I assume low-level chronic exposure to H₂S will also provide resistance to hypoxia.
2. Further, how mechanistically explain the lack of response in the heart tissue, which is also sensitive to hypoxia injury? SPC in mice caused a significant increase in SQOR in the brain, but not in the heart. The selective SPC effect to the brain is a needed feature in all of the experimentations shown here, but the underlying mechanism is very important, and which was not examined and explained in the current study. Why is SQOR in the heart not responsive to H₂S SPC?
3. Fig 1N: SPC in mice caused a significant increase in SQOR in the brain, but not in the heart. I assume the two columns of gel blots in control and SPC, as shown in Fig 1N represent repeats of two mice samples. Please clarify and label them accordingly. The level of SQOR increases in SPC group is really marginal. In contrast, CBS in SPC appears to be significantly down-regulated. A better job has to be done to present time- and dose-dependency of SQOR expression throughout the pre-conditioning process.

4. What's even more striking is that chronic inhaling of H₂S in the control group showed an increased level of sulfide and thiosulfate in the plasma and the brain, while the pre-conditioning group showed a similar level only in the plasma, but not in the brain (Fig 1E, F). Authors argued that pre-conditioned mice are resistant to hypoxia injury because of the upregulated sulfide catabolism in the brain. The snapshot of this, as shown in Fig 1E and F, needs to be expanded to include a time course of what the critical time point was for the upregulation of the sulfide catabolism in the brain.

5. Line 143: as stated by the authors and the quoted references, breathing H₂S depresses metabolism and decreases the body temperature of rodents. As we know that lowering body temperature significantly affects the outcomes of cerebral ischemia in rodents, it is therefore critical to show measurements of body temp changes in chronic inhalation of H₂S and SPC mice. Whether this temperature changes lead to tolerance to hypoxia.

6. Fig 6K bilateral occlusion of carotid arteries should be called 2VO rather than BCAO. H&E staining of BCAO brain hippocampus, as shown in Fig 6K are of poor quality. Cell death (neurons or other cell types) appeared to occur in CA2, CA1. Additional staining is required to show cell death and protection, such as the Fluoro-Jade stain. What happened to neurons in the cortex areas, are they protected?

7. Along the same vein as above, what happened to the vasculatures and blood flow during the treatment? Are these altered and contributed to the protection? Data need to be shown. In the MCAO studies, pharmacological scavenging of sulfide using SS-20 was examined at 48 h after reperfusion (line 612). But I thought this is a permanent MCAO. Why reperfusion? Further, authors seem to use 2VO, pMCAO, and tMCAO in three different experiments. What are the underlying connections amongst these three hypoxia injury models? Changes in body temperature and vasculature and blood flow should be determined.

8. Fig S1H: The expression of HIF-1a target genes at 24 h after the last H₂S exposure did not show any differences compared with the controls. Because of the transient and inducible nature of HIF-1a expression, it is necessary to show a time course expression of these target genes during early exposure to H₂S in order to rule out HIF1a involvement.

9. Fig 1M, O, relative gene expression to what? Panel N: The two columns of bands of Control and SPC should be labeled. It is not convincing based on panel N to conclude a robust increase in SQOR. No blots were shown for the heart (P). It is essential to show a time point change of SQOR throughout the pre-conditioning process.

10. Fig 2A: representative PCR gels need to be shown. Further, comparisons in mRNA and protein levels between male and female rats and hypoxia-tolerant ground squirrels need to be demonstrated.

11. Fig 2D, E: it is known that estrogen has a profound protective effect on resistance to hypoxia. Panel D should show both mRNA and proteins expressions levels. It is interesting to see that ovariectomized females have reduced SQOR. How long and the dosage of estrogen supplement used should be also be shown. SQOR is regulated by estrogen in females. Is this true in male mice and the ground squirrels?

12. Fig 2L representative blots and gels should be shown. In this experiment, AAV knock down the expression of SQOR was designed and used. It would also be interesting to show

the effects of using a pharmacological inhibitor to SQOR.

13. Fig 3D is the N-terminal deleted mutant mice Western blot. I assume the antibody should pick up a shorter band in the Sqor mice, although it could be a minimal shift in size. So, the current top panel of D is not convincing.

14. Fig 3E, H: Line 332-335: OGD experiment to show sulfide level in wt and mutant mice. Data need to be shown in cell viability. How was it assessed? Furthermore, line 335 said SQOR has a critical role in the catabolism of sulfide in neurons even under normoxic conditions. This contradicts the statement in Line 339, which said, "When Sqor Δ N/ Δ N and control mice breathed air, there were no differences in the brain levels of sulfide and persulfide or in the ratio of NADH/NAD⁺". Explain?

15. Are mutant mice resistant to BCAA, MCAO?

16. Line 386: "Brain sulfide levels were comparable between rats and 13LG squirrels breathing room air." Needs data to support the statement.

17. Statistics need to be done with the presentation of the exact p-value and df shown where appropriate.

18. The method for tMCAO (Line 1365) needs a better description. Did you measure the blood flow? Do Sqor Δ N/ Δ N mice have the same vasculature and blood flow as the WT?

19. The authors need to state whether the experimental design followed the ARRIVE guidelines. What are the inclusion and exclusion criteria used and how randomization was used in the animal studies?

Reviewer #2 (Remarks to the Author):

Marutani and colleagues investigated in this work an unexpected role of sulfide in hypoxia tolerance. They initially found that intermittent exposure of mice to hydrogen sulfide induced an unexpected tolerance to severe hypoxia, possibly by inducing the sulfide-quinone oxidoreductase (SQOR) in the brain. No changes in HIF1 α -dependent pathways were observed, excluding this as a possible explanation of the hypoxia resistant phenotype. Interestingly, females were less affected, and the authors clearly demonstrate by a number of approaches that hormones are responsible for this effect.

Next, they investigated the effects of preventing SQOR to go to mitochondria by deleting the mitochondrial targeting signal in mice. SQOR Δ N animals showed a severe phenotype and a markedly increased sensitivity to hypoxia.

Interestingly, the authors also found that ground squirrels, which are naturally resistant to hypoxia, have extremely high levels of sulfide in the brain, compared to rats. Accordingly, downregulation of SQOR led to accumulation of NADH, sulfide and persulfide, as the authors found in mice exposed to NaS.

Then, the authors studied the effects of NaS on SH-SY5Y cells, which have very low SQOR levels. Overexpression of SQOR caused accumulation of persulfides in 21% oxygen due to increased sulfite consumption and showed increased ATP production in mitochondria when exposed to NaS, suggesting that in normoxic conditions SQOR contributes to the OXPHOS. However, in hypoxic conditions SQOR expression prevented the accumulation of sulfide,

ROS and lactate and prevented the increase of NADH/NAD ratio. In addition, while in control cells ATP production was reduced by COX inhibition, cells overexpressing SQOR were protected.

Finally, the authors found that brain-specific overexpression of SQOR prevents ischemic brain injury by blunting hypoxic damage.

The paper is interesting, but I have several comments that the authors should consider.

Major concerns

1) The authors use NaS, a sulfide donor, to mimic hypoxia in cells, but I am not convinced that this approach can be considered as analogous to hypoxia. The hypoxic response is well known and involves HIF1alpha, which is not induced by sulfide preconditioning.

2) Using 5% oxygen seems rather extreme, and the reason for using this concentration is unclear. A similar comment can apply to the amount of sulfide during preconditioning, and to the concentration of NaS (page 9).

3) Why did the authors decide to delete SQOR only in mitochondria and not just doing a knockout? Is there a non-mitochondrial SQOR? Is residual SQOR active in the cytosol? Can the authors exclude that the phenotype is due to a toxic effect of misplaced SQOR? Did they observe any phenotype in the heterozygous mice? No information is given on this matter.

4) On page 12, the authors report that mitochondrial SQOR prevents accumulation of sulfide and persulfide not only in brain but also in heart and liver. However, no information is given on these organs before. How are SQOR levels in heart and liver? How are sulfide levels?

5) The story on the ground squirrel is interesting, but I am not sure it fits with the whole paper. And I am not convinced the rat is an appropriate control. In addition, why succinate accumulates in the brains of 13LG squirrels is not explained.

Minor points

1) The order of the panels in Figure 1 is not very logical

2) Figure S1: some characters are too small (in particular the p values).

3) Why sulfide levels are upregulated in hypoxia, as reported in the discussion, is unclear to this reviewer.

Reviewer #3 (Remarks to the Author):

Marutani et al. report on a series of interesting and original findings indicating that levels of the mitochondrial sulfide-catabolizing enzyme sulfide:quinone oxidoreductase (SQOR) determine vulnerability of the brain to hypoxia. Starting with the discovery that H₂S preconditioning confers tolerance to hypoxia by increasing brain sulfide catabolism, they convincingly show that upregulation and mitochondrial localization of SQOR underlies this effect. Studies with isolated mitochondria further suggest that sulfides directly inhibit ETC complexes, providing a possible explanation for COX4 inhibition during hypoxia despite the high oxygen affinity of the enzyme. Other novel findings include sexual dimorphic expression of SQOR in CD1 mice that may explain greater tolerance of female mice for hypoxia. As well, the group makes a compelling argument that the well-documented hypoxia tolerance of 13-lined ground (13LG) squirrels may result from elevated SQOR levels compared to other rodents. SQOR overexpression experiments in mice and a human neuroblastoma cell line further support a neuroprotective role of sulfide catabolism. Finally, sulfide scavenging in global and focal brain ischemia models underscore the clinical relevance of the report. In light of the novelty of the findings and the comprehensiveness of the experiments,

this study is likely to have a major impact in the field and to appeal to the journal's diverse readership. There are, however, some concerns that should be addressed.

1) From Fig. 3D, it is not clear whether SQOR levels are decreased in SqrDN/DN mice. A possible decrease in stability of the protein lacking the mitochondrial leader sequence would impact the interpretation of these experiments. Please provide a quantification.

2) An obvious question from studies with the 13LG squirrel model is whether AAV-shSQOR impacts vulnerability to hypoxia. The relatively moderate effects on sulfide catabolism and NADH/NAD⁺ ratios suggest that this may not be the case.

3) What explains the increased persulfide levels in SPC mice breathing 21% O₂ compared to control mice (Fig. 1J)?

4) The concluding sentence in l. 157,158 is too strongly worded this early in the manuscript.

5) L. 193 ("Chronic exposure ..") is missing references.

Signed:

Stefan Strack

REVIEWER COMMENTS

Reviewer #1 (Remarks to the Author):

Hydrogen sulfide is a member of a small family of labile biological signaling gasotransmitters. H₂S plays a broad range of physiological and pathophysiological functions, including induction of angiogenesis, regulation of neuronal activity, vascular relaxation, glucose homeostatic regulation, and protection against I/R injury in the heart, liver, kidney, lung, and brain.

In this MS, authors serendipitously found intermittent breathing in H₂S for 5 days pre-conditioned mice against cerebral ischemic insult. SPC mice will not show hypo-metabolic effects than the acute and chronic inhalation of H₂S (control). Chronic exposure to H₂S is lethal. Female mice showed a higher degree of resistance to hypoxia since it has a higher level of SQOR in the brain, while mice lacking the expression of mitochondrial SQOR showed higher sensitivity to hypoxia. Comparisons between hypoxia-tolerant ground squirrels and rats or mice were also made to highlight the higher capacity to catabolize sulfide in the brain of ground squirrels since they express higher levels of SQOR compared to mice and rats. SSQOR knocking down using AAV shRNA reduced ground squirrels' resistance to hypoxia. The authors went on to show SQOR is responsible for ischemic brain injury in two cerebral ischemia models: 2VO (BCAO) and permanent pMCAO.

These findings are interesting, and some are surprising. Intermittent inhalation of H₂S is protective with a pre-conditioning effect is novel, so is the deletion of SQOR studies. The MS is written in a logical manner. The strength of the manuscript is that multiple approaches were used to determine the role of SQOR in H₂S catabolism and its relevance to hypoxia tolerance in several cerebral ischemia models. However, parts of the results are just too brief, and not enough details to support the statement made.

Major concerns are:

1. The lack of investigation on the potential mechanisms between chronic exposure to H₂S and intermittent chronic exposure to H₂S. What if a chronic, but non-lethal low dose treatment with H₂S was used. Does that provide pre-conditioning towards hypoxia tolerance? Increased tolerance to H₂S cannot only be assessed using lethality as a sole or major criterion. I assume low-level chronic exposure to H₂S will also provide resistance to hypoxia.

We did not assess the tolerance to H₂S by lethality to chronic H₂S inhalation in this manuscript. We unexpectedly found that intermittent inhalation of H₂S induces tolerance to hypoxia in mice that was assessed using lethality in hypoxia, but not H₂S. Upon examining the mechanisms responsible for the hypoxia tolerance induced by sulfide pre-conditioning, we discovered that upregulation of SQOR and sulfide catabolism underscore hypoxia tolerance of the brain. Therefore, the goal of this manuscript is to elucidate the role of sulfide catabolism in hypoxia tolerance. Increased tolerance to hypoxia was assessed not only using lethality in low oxygen environment but also using multiple models of cerebral ischemia. While it is potentially interesting to examine the effects of different protocols (i.e., dose and duration) of sulfide pre-

conditioning on resistance to hypoxia, we believe such studies are beyond the scope of the current investigation.

2. Further, how mechanistically explain the lack of response in the heart tissue, which is also sensitive to hypoxia injury? SPC in mice caused a significant increase in SQOR in the brain, but not in the heart. The selective SPC effect to the brain is a needed feature in all of the experimentations shown here, but the underlying mechanism is very important, and which was not examined and explained in the current study. Why is SQOR in the heart not responsive to H₂S SPC?

To explain the lack of SPC-induced increase in SQOR levels in the heart tissue, we compared SQOR levels in the brain and heart. We observed that heart tissue has approximately 10-fold higher SQOR levels at baseline than in the brain tissue in mice. While SPC increased the SQOR levels in the brain, it did not increase SQOR levels in the heart. These observations suggest that in the brain, the basal level of SQOR is insufficient to catabolize H₂S during H₂S inhalation, thus repetitive H₂S inhalation induced upregulation of SQOR to prevent H₂S accumulation in the brain. In contrast, there is more than sufficient level of SQOR expressed at baseline in the heart to catabolize H₂S provided by intermittent inhalation of 80 ppm H₂S. Therefore, sulfide does not accumulate in the heart after repeated H₂S inhalation and upregulation of SQOR was not induced. These results are furnished as Figure 1, O and P, Figure S1D and in the Results as follows:

“Acquired tolerance to acute hypoxia in sulfide pre-conditioned mice was associated with increased levels of SQOR in the brain, but not in the heart presumably due to higher baseline SQOR levels in the heart than in the brain (**Fig. 1, M-O, Fig. S1D**).” (Page 6, 3rd paragraph). In addition, our observation in the study of *Sqor*^{AN/AN} mice and their wild-type littermates support the hypothesis that the higher native levels of SQOR in the heart is required and sufficient to block hypoxia-induced increase of sulfide in the heart, thus corroborating the lack of SQOR upregulation in the heart after sulfide pre-conditioning. To address this point, we added the following paragraph in the Discussion:

“These observations suggest that basal levels of SQOR expression in heart and liver, but not in brain, are sufficient to catabolize sulfides and maintain oxidative phosphorylation when breathing 5.5% O₂ for 3 min. This hypothesis is corroborated by the lack of SQOR upregulation in the heart by sulfide pre-conditioning; basal SQOR levels in the heart are sufficient to catabolize sulfide during intermittent H₂S inhalation.” (Page 33, 2nd paragraph).

3. Fig 1N: SPC in mice caused a significant increase in SQOR in the brain, but not in the heart. I assume the two columns of gel blots in control and SPC, as shown in Fig 1N represent repeats of two mice samples. Please clarify and label them accordingly. The level of SQOR increases in SPC group is really marginal. In contrast, CBS in SPC appears to be significantly down-regulated. A better job has to be done to present time- and dose-dependency of SQOR expression throughout the pre-conditioning process.

To address reviewer’s concern, we omitted original Figure 1N and replaced it with immunoblots of SQOR in the brain and heart of control and SPC mice as presented in Fig 1, N and O and additional blots presented in Fig S1, D and F. Immunoblots for other sulfide metabolizing enzymes in the brain and heart are presented in Fig S2 B and C. Analysis of brain CBS

immunoblots presented in Fig S2B shows non-significant trend of increase in the brain CBS levels after SPC. All uncropped immunoblots are submitted in a separate file. To address the time-dependency of SQOR expression during SPC, we measured brain SQOR levels after 2 or 5 days of SPC. We observed that SPC for 5 days, but not 2 days, increased brain SQOR levels and induced hypoxia tolerance in mice. These results are now presented in Figure S1 E and F and in Results as follows:

“Increments of brain SQOR levels by sulfide pre-conditioning temporarily coincided with the acquisition of hypoxia tolerance; sulfide pre-conditioning for 5 days, but not 2 days, induced hypoxia tolerance and increased brain SQOR levels (**Fig. S1, E and F**).” (Page 6, 3rd paragraph).

4. What’s even more striking is that chronic inhaling of H₂S in the control group showed an increased level of sulfide and thiosulfate in the plasma and the brain, while the pre-conditioning group showed a similar level only in the plasma, but not in the brain (Fig 1E, F). Authors argued that pre-conditioned mice are resistant to hypoxia injury because of the upregulated sulfide catabolism in the brain. The snapshot of this, as shown in Fig 1E and F, needs to be expanded to include a time course of what the critical time point was for the upregulation of the sulfide catabolism in the brain.

Please see our response to Comments 3.

5. Line 143: as stated by the authors and the quoted references, breathing H₂S depresses metabolism and decreases the body temperature of rodents. As we know that lowering body temperature significantly affects the outcomes of cerebral ischemia in rodents, it is therefore critical to show measurements of body temp changes in chronic inhalation of H₂S and SPC mice. Whether this temperature changes lead to tolerance to hypoxia.

As shown by others and us, hypo-metabolic effects of H₂S breathing in rodents is short-lived and quickly reversed within 10-20 min when H₂S breathing is stopped (Blackstone et al., 2005; Volpato et al., 2008). Because we exposed SPC-treated mice to hypoxia 24 hours after the last H₂S breathing session, body temperature of mice completely returned to normal as shown as “D6-pre” in Figure 1A when mice were exposed to hypoxia. This important point is described in the Results as follows:

“Mice were studied twenty-four hours after the last H₂S exposure (on the 6th day), when the metabolism and body temperature of the mice had completely recovered (**Fig. S1A**).” (Page 5, last paragraph)

In addition, we maintained mice at 37°C when mice were anesthetized and ventilated with hypoxic gas to harvest brain tissues. As shown in Fig 1 K and L, SPC prevented the hypoxia-induced increase of NADH/NAD ratio and lactate levels in brain tissues obtained at 37°C. This information was added to the Results as follows:

“To examine the impact of sulfide pre-conditioning on biochemical changes in the brain during hypoxia, mice were anesthetized with isoflurane and ventilated with 21% or 5% O₂ for 3 min at 37°C and brains were harvested and snap frozen in liquid nitrogen.” (Page 6, 2nd paragraph)
Taken together, it is highly unlikely that H₂S-induced hypothermia contributed to hypoxia tolerance of the mouse brain shown in this study.

6. Fig 6K bilateral occlusion of carotid arteries should be called 2VO rather than BCAA. H&E

staining of BCAO brain hippocampus, as shown in Fig 6K are of poor quality. Cell death (neurons or other cell types) appeared to occur in CA2, CA1. Additional staining is required to show cell death and protection, such as the Fluoro-Jade stain. What happened to neurons in the cortex areas, are they protected?

BCAO was replaced with 2VO throughout the manuscript. We stained additional brain sections of mice infected with AAV-SQOR or AAV-GFP and subjected to 2VO with Fluoro-Jade B to detect cell death in hippocampus and cortex areas. In these sections, we found that AAV-mediated SQOR expression prevented cell death detected by FJB in the hippocampus and cortex of mice. Representative brain sections stained with FJB with quantification of FJB positive neurons are now presented in Figure 6 K-O. Original Figure 6K of H&E-stained brain sections with quantification of viable neurons are now presented in Figure S9. Accordingly, Results were revised as follows:

“In control mice that were infected with AAV-GFP and subjected to 2VO and reperfusion, a number of Fluoro Jade B (FJB)-positive dead neurons were observed in the hippocampal CA1 and CA3 regions (**Fig. 6, K-M**) and cerebral cortex (**Fig. 6, N and O**). In mice expressing SQOR in neurons, only a few FJB-positive neurons were found in hippocampus and cortex after 2VO and reperfusion. The number of viable neurons counted in H&E-stained brain sections showed a reciprocal trend of the number of FJB-positive dead neurons in these brain regions (**Fig. S9**).” (Page 24, 3rd paragraph)

7. Along the same vein as above, what happened to the vasculatures and blood flow during the treatment? Are these altered and contributed to the protection? Data need to be shown. In the MCAO studies, pharmacological scavenging of sulfide using SS-20 was examined at 48 h after reperfusion (line 612). But I thought this is a permanent MCAO. Why reperfusion? Further, authors seem to use 2VO, pMCAO, and tMCAO in three different experiments. What are the underlying connections amongst these three hypoxia injury models? Changes in body temperature and vasculature and blood flow should be determined.

We measured relative cerebral blood flow changes during and after transient MCAO in wild-type mice treated with sulfide scavenger SS-20 or vehicle. We observed that scavenging sulfide did not affect cerebral blood flow during or after transient MCAO. These results suggest that scavenging sulfide does not affect cerebral blood flow during and immediately after cerebral ischemia. Further, in combination with our observation that sulfide preconditioning did not induce VEGF, these observations indicate that neuroprotective effects of enhanced sulfide catabolism against cerebral ischemia are unlikely to be mediated by increasing CBF and/or promoting cerebral angiogenesis. These results are presented in Figure S10D and Results and Discussion were revised as follows:

“Treatment with SS-20 during ischemia markedly decreased the infarct size and improved neurological function at 48h after reperfusion without changing CBF (Fig. S10).” (Page 29, last paragraph)

“Sulfide scavenging did not affect CBF during and after cerebral ischemia and sulfide preconditioning did not induce VEGF in the brain. These observations suggest that beneficial effects of sulfide scavenging or SQOR during hypoxia are unlikely to be mediated by increasing CBF or promoting cerebral angiogenesis.” (Page 34, 3rd paragraph)

Because we overexpressed SQOR by injecting AAV-SQOR bilaterally in newborn mice, we initially performed 2VO model to determine the magnitude and spatial distribution of SQOR overexpression and cerebral protection in whole brain areas. We then used permanent MCAO model to quantitate the effects of sulfide scavenging on ischemic injury because accurate quantification of the volume of brain injury in 2VO model is more challenging than in pMCAO model. Lastly, we examined effects of sulfide scavenging with SS-20 on the brain injury induced by transient MCAO to address effects of sulfide scavenging on cerebral blood flow during and after cerebral ischemia and because tMCAO is considered to be more clinically relevant than pMCAO.

Please see our response to Comment #5 about body temperature.

8. Fig SIH: The expression of HIF-1 α target genes at 24 h after the last H₂S exposure did not show any differences compared with the controls. Because of the transient and inducible nature of HIF-1 α expression, it is necessary to show a time course expression of these target genes during early exposure to H₂S in order to rule out HIF1 α involvement.

As the reviewer pointed out, HIF-1 α expression is transient. While it is possible that HIF-1 α and its canonical targets may be upregulated during or early after H₂S breathing, our results showed that canonical HIF-1 α targets were not upregulated when mice were exposed to hypoxia at 24h after 5 days of sulfide pre-conditioning. Thus, we believe our results rule out the involvement of the canonical HIF-1 α targets to the protective effects of sulfide pre-conditioning during hypoxia. Nonetheless, our results do not completely rule out the role of HIF-1 α *per se* (not via canonical targets) in the current study. To acknowledge this important point, we added sentences in Discussion as follows:

“Although we did not find upregulation of canonical HIF-1 α targets in the mouse brain when mice were exposed to hypoxia at 24h after the last H₂S inhalation, the role of HIF-1 α in the induction of hypoxia tolerance of sulfide-preconditioned mice remains incompletely defined. H₂S has been reported to inhibit (Kai et al., 2012) or activate HIF-1 α (Budde and Roth, 2010). It is possible that sulfide pre-conditioning induces SQOR via HIF-1 α activation. Interaction between HIF-1 α and sulfide oxidation pathway in hypoxia tolerance remains to be elucidated in future studies.” (Page 34, last paragraph)

9. Fig 1M, O, relative gene expression to what? Panel N: The two columns of bands of Control and SPC should be labeled. It is not convincing based on panel N to conclude a robust increase in SQOR. No blots were shown for the heart (P). It is essential to show a time point change of SQOR throughout the pre-conditioning process.

Figure 1M shows the relative SQOR mRNA levels of the brain of control and SPC mice measured by real-time qPCR and reported as relative SQOR mRNA levels of control mice as 1. Figure 1O shows relative brain SQOR protein levels as brain SQOR protein levels (calculated as a ratio to loading control) of control mice as 1. Original Figure 1N was replaced and blots for the heart are shown in Figure 1O as explained in our response to comment #3. To further address reviewer's concern, additional representative brain SQOR immunoblots are presented in Figure

S1 D and F. In addition, all uncropped immunoblots used in this manuscript are now submitted as a separate file. Please see our response to your comment #2 regarding time-dependent changes of SQOR expression levels (Figure S1 E and F).

10. Fig 2A: representative PCR gels need to be shown. Further, comparisons in mRNA and protein levels between male and female rats and hypoxia-tolerant ground squirrels need to be demonstrated.

Because we performed a real-time qPCR, there is no PCR gels to be shown. However, immunoblots are shown in 2B. Although comparison of SQOR levels between male and female rats and ground squirrels is of interest, ground squirrels are not available year-round and their brain tissues cannot be obtained until next late spring/summer. To avoid further delay of publication of this manuscript, we decided not to perform these experiments.

11. Fig 2D, E: it is known that estrogen has a profound protective effect on resistance to hypoxia. Panel D should show both mRNA and proteins expressions levels. It is interesting to see that ovariectomized females have reduced SQOR. How long and the dosage of estrogen supplement used should be also be shown. SQOR is regulated by estrogen in females. Is this true in male mice and the ground squirrels?

In response to the reviewer's request, we attempted to detect SQOR protein levels with immunoblots with the remaining brain samples. We found a trend of brain SQOR protein levels that parallels that of SQOR mRNA; ovariectomy decreased SQOR levels while estrogen supplementation tended to restore SQOR levels in the brain, although statistically significant difference was not found between ovariectomized mice with or without estrogen supplementation in part due to a smaller sample size ($p=0.170$). These results are now presented as Figure 2E and text was revised accordingly as follows:

“Of note, higher SQOR levels in the brain of female mice appears to be estrogen dependent. Ovariectomy decreased mRNA and protein levels of SQOR in the brain and abolished hypoxia tolerance in female mice, whereas estrogen supplementation restored brain SQOR mRNA levels and hypoxia tolerance and tended to restore SQOR protein levels in ovariectomized female mice (**Fig. 2, D-F**).” (Page 13, 1st paragraph).

For estrogen supplementation, silicon capsule filled with a sesame oil with or without 17 β -estradiol (Sigma-Aldrich, E8875) at 36 μ g/ml was implanted subcutaneously on the back of mice, as reported previously.⁶¹ Mice were used for experiments after 2 weeks of estrogen supplementation. This is furnished in Methods (Page 48, 2nd paragraph). We believe effects of estrogen in male mice and 13LGS are beyond the scope of this study.

12. Fig 2L representative blots and gels should be shown. In this experiment, AAV knock down the expression of SQOR was designed and used. It would also be interesting to show the effects of using a pharmacological inhibitor to SQOR.

Due to the lack of remaining brain tissues and difficulty in synthesizing a new batch of AAV vector to conduct additional knockdown experiments during the lab and vector core lockdown during this pandemic, we could not obtain immunoblots for Figure 2L. To the best of our

knowledge, there is no pharmacological SQOR inhibitor available on the market.

13. Fig 3D is the N-terminal deleted mutant mice Western blot. I assume the antibody should pick up a shorter band in the *Sqor* mice, although it could be a minimal shift in size. So, the current top panel of D is not convincing.

Most mitochondrial proteins such as SQOR contain an N-terminal targeting signal that is removed proteolytically by a specific protease following import into mitochondria. We used online tool Mitoprot II to calculate mitochondrial targeting sequence and the cleavage site of SQOR protein. The putative cleavage site is Val67 in SQOR protein, and the final mature SQOR protein consisting of 383 amino acids lacks the N-terminal region 1-67. Illegitimate translation can be initiated from an in-frame ATG other than the authentic translation initiation codon in genome edited animal. In *Sqor*^{ΔN/ΔN} mice, the 14 bp including first ATG in exon 2 of *Sqor* is deleted, and the second methionine of SQOR positioned at 58 becomes the translational start site. This results in a 393 amino acids product of SQORΔN utilizing the next in-frame methionine as the translation start site. Taken together, the lengths of wild-type mature SQOR protein and SQORΔN protein are almost identical, which makes it difficult to distinguish the faint difference in size (or electrophoretic mobility) of wild-type and SQORΔN proteins on the western blotting.

14. Fig 3E, H: Line 332-335: OGD experiment to show sulfide level in wt and mutant mice. Data need to be shown in cell viability. How was it assessed? Furthermore, line 335 said SQOR has a critical role in the catabolism of sulfide in neurons even under normoxic conditions. This contradicts the statement in Line 339, which said, "When *Sqor*ΔN/ΔN and control mice breathed air, there were no differences in the brain levels of sulfide and persulfide or in the ratio of NADH/NAD⁺". Explain?

Cell viability was assessed with crystal violet assay and the results are presented in Figure S6B and results were revised as follows:

"Reoxygenation after OGD decreased cell viability more markedly in *Sqor*^{ΔN/ΔN} than in WT neurons (**Fig. S6B**)."

 (Page 15, last sentence)

While exclusion of SQOR from mitochondria increased sulfide levels in primary neurons cultured in air (Fig. 3H), SQOR deficiency in mitochondria did not affect sulfide and NADH/NAD⁺ in air breathing mice brain (Fig. 3 J and K). This apparent difference is likely due to the difference in tissue samples actually used; primary cultured neurons vs whole brain tissue homogenates. While intracellular H₂S levels were measured in primary cultured neurons, brain levels of H₂S were measured in whole brain tissue homogenates presumably containing mixed cell types. In addition, a significantly longer time is required to obtain brain tissue homogenates than to obtain cell lysates for the measurements of H₂S even in the hands of experienced experimenter. Because H₂S is volatile and rapidly lost into the atmosphere, the longer preparation time may have diminished the H₂S signal to the point that the detection of relatively small difference is no longer possible. To clarify this point, we revised the sentence as follows: "When *Sqor*ΔN/ΔN and control mice breathed air, there were no differences in the levels of sulfide and persulfide or in the ratio of NADH/NAD⁺ in whole brain tissue homogenates." (Page 16, 1st paragraph)

15. Are mutant mice resistant to BCAA, MCAO?

Sqor^{ΔN/ΔN} mice are likely to be more sensitive to BCAA or MCAO but it is not possible to do either model as the *Sqor*^{ΔN/ΔN} mice are too small.

16. Line 386: “Brain sulfide levels were comparable between rats and 13LG squirrels breathing room air.” Needs data to support the statement.

Brain sulfide levels of rats and 13LG squirrels breathing air or 5% oxygen are shown in Figure 4B.

17. Statistics need to be done with the presentation of the exact p-value and df shown where appropriate.

After consulting a number of published papers in Nature Communications, we chose to present the results of statistics with symbols (*) rather than exact p-values to improve legibility of multi-panel figures. However, if editors prefer exact p-values for all figures, we are happy to provide them.

18. The method for tMCAO (Line 1365) needs a better description. Did you measure the blood flow? Do *Sqor*^{ΔN/ΔN} mice have the same vasculature and blood flow as the WT?

The method for tMCAO was revised to include more details including the measurements of CBF during tMCAO as follows:

“Transient middle cerebral artery occlusion (tMCAO) of mice—WT CD-1 mice (male, 8-10 weeks old) were subjected to transient MCAO for 60 min and reperfusion for 48h according to a previous report with some modification (Shvedova et al., 2019). Briefly, mice were anesthetized with 1.5% isoflurane in oxygen. Mice breathed 100% oxygen with spontaneous breathing from the induction of anesthesia to 15 min after MCA reperfusion. Rectal temperature was monitored and kept at 37±0.5°C using a heating pad under mice during surgery until mice recover from anesthesia. Changes in cerebral blood flow (CBF) was monitored during surgery to confirm the occlusion of MCA using a laser Doppler flowmetry (Moor instruments, Inc.) probe positioned at 2 mm posterior, 5 mm lateral from the Bregma. Common carotid artery (CCA) was occluded with a microclip and a 7-0 monofilament was inserted via the isolated external carotid artery (ECA) through an internal carotid artery to occlude MCA. MCA was reperfused by removing a monofilament at 60 min after starting MCAO. CCA was reperfused after ECA closure by removing a microclip at 15 min after MCA reperfusion. Saline or SS20 at 250 μmol/kg was administered IV at 15 min before MCA reperfusion. Taking circadian rhythm into account, all mice were subjected to surgery in the daytime (9:00 to 17:00). Mice received 1 ml of 5 % dextrose-enriched lactated Ringer’s solution (ip) daily to avoid dehydration. After evaluating neurological functional score (Desland et al., 2014), brains were harvested to measure cerebral infarct volume using TTC staining at 48h after MCAO.”(Page 50, 3rd paragraph).

Please also see our response to your comment #7 regarding cerebral blood flow. *Sqor*^{ΔN/ΔN} mice showed growth retardation in whole body as we described in main text and figures. This phenotype hindered us in appropriately analyzing the phenotypes in each organ, including vasculatures. Meanwhile, we investigated the heart morphology with the electron microscopic analysis and no particular phenotype or pathological changes were found in the

heart of *Sqor*^{ΔN/ΔN} mice. The mutant mice, therefore, unlikely have severe phenotypes in heart. Nevertheless, more extensive investigations may be necessary to gain further understanding of SQOR functions in other organs like the vascular system.

19. The authors need to state whether the experimental design followed the ARRIVE guidelines. What are the inclusion and exclusion criteria used and how randomization was used in the animal studies?

Following statement was included in the Method section.

“The design of experiments involving animal subjects followed the ARRIVE guidelines. To minimize variability, we used randomized paired (a.k.a. matched pairs) design. We paired mice to two or more treatment groups on the basis of similar weight, age, delivery date, and when possible holding cage.” (Page 35, 4th paragraph)

No animal was excluded in this study.

This information is also included in the Reporting Summary document.

Reviewer #2 (Remarks to the Author):

Marutani and colleagues investigated in this work an unexpected role of sulfide in hypoxia tolerance. They initially found that intermittent exposure of mice to hydrogen sulfide induced an unexpected tolerance to severe hypoxia, possibly by inducing the sulfide-quinone oxidoreductase (SQOR) in the brain. No changes in HIF1alpha-dependent pathways were observed, excluding this as a possible explanation of the hypoxia resistant phenotype. Interestingly, females were less affected, and the authors clearly demonstrate by a number of approaches that hormones are responsible for this effect.

Next, they investigated the effects of preventing SQOR to go to mitochondria by deleting the mitochondrial targeting signal in mice. SQOR ΔN animals showed a severe phenotype and a markedly increased sensitivity to hypoxia.

Interestingly, the authors also found that ground squirrels, which are naturally resistant to hypoxia, have extremely high levels of sulfide in the brain, compared to rats. Accordingly, downregulation of SQOR led to accumulation of NADH, sulfide and persulfide, as the authors found in mice exposed to Na₂S.

Then, the authors studied the effects of Na₂S on SH-SY5Y cells, which have very low SQOR levels. Overexpression of SQOR caused accumulation of persulfides in 21% oxygen due to increased sulfite consumption and showed increased ATP production in mitochondria when exposed to Na₂S, suggesting that in normoxic conditions SQOR contributes to the OXPHOS. However, in hypoxic conditions SQOR expression prevented the accumulation of sulfide, ROS and lactate and prevented the increase of NADH/NAD ratio. In addition, while in control cells ATP production was reduced by COX inhibition, cells overexpressing SQOR were protected. Finally, the authors found that brain-specific overexpression of SQOR prevents ischemic brain injury by blunting hypoxic damage.

The paper is interesting, but I have several comments that the authors should consider.

Major concerns

1) The authors use Na₂S, a sulfide donor, to mimic hypoxia in cells, but I am not convinced that

this approach can be considered as analogous to hypoxia. The hypoxic response is well known and involves HIF1alpha, which is not induced by sulfide preconditioning.

Because acute hypoxia increases H₂S levels in tissues with low levels of SQOR including brain and H₂S is known to inhibit mitochondrial respiration, we hypothesized that sulfide preconditioning upregulates the ability of brain mitochondria to catabolize sulfide and thereby prevents sulfide-induced inhibition of oxidative phosphorylation. To address this hypothesis, we used Na₂S to mimic effects of hypoxia-induced increase of intracellular sulfide levels on mitochondrial respiration in isolated brain mitochondria. Effects of hypoxia were examined in vivo and in vitro in other experiments in this study and their results are in line with the results of experiments with Na₂S in isolated brain mitochondria. Therefore, we believe the use of Na₂S in the isolated mitochondria experiments as substitute of hypoxia is justifiable. Please see our response to Reviewer 1 Comment #8 regarding the role of HIF1a in sulfide preconditioning.

2) Using 5% oxygen seems rather extreme, and the reason for using this concentration is unclear. A similar comment can apply to the amount of sulfide during pre-conditioning, and to the concentration of Na₂S (page 9).

We used 5% oxygen in hypoxia breathing studies because exposure to 5% or lower oxygen concentration has been used to test hypoxia tolerance in rodents in previous studies by others (D'Alecy et al., 1990; Park et al., 2017; Stobdan et al., 2018). Concentration of H₂S used for preconditioning was 80 ppm which was used in our previous study and found to induce tolerance to H₂S (Kida et al., 2014). We adopted the amount of Na₂S that was found to affect mitochondrial respiration in previous studies by others and us (Ikeda et al., 2016; Szabo et al., 2014).

3) Why did the authors decided to delete SQOR only in mitochondria and not just doing a knockout? Is there a non-mitochondrial SQOR? Is residual SQOR active in the cytosol? Can the authors exclude that the phenotype is due to a toxic effect of misplaced SQOR? Did they observe any phenotype in the heterozygous mice? No information is given on this matter.

We had tried to make simple knock-out mice of *Sqor*, but failed to generate the mutant mice. This result implies that SQOR is essential for embryogenesis. It prompted us to take strategy to generate mitochondria specific knock-out mice of *Sqor*. *Sqor*^{ΔN/+} mice which one of *Sqor* allele expresses SQOR in cytosol showed no phenotypes under normal physiological conditions over one year, as shown in Figure 3, E-G. In addition, the metabolome analysis of MEFs derived from *Sqor*^{ΔN/ΔN} mice revealed no major changes in the dynamic profile of cytosol metabolites, except for the severe impairment of the mitochondrial energy metabolism. These data support the notion that the phenotypes of *Sqor*^{ΔN/ΔN} mice are entirely due to complete elimination of the SQOR protein in mitochondria and excludes the possibility of toxic effects of the SQOR mislocalization in cytosol. Ubiquinone, a co-enzyme of SQOR is found in not only mitochondria but also cytosol. It is likely that SQOR may be functionally active in terms of sulfide metabolism, which might somehow compensate for the impaired mitochondrial sulfur metabolism. This putative functions of SQOR localized or remaining in the cytosol may explain relatively milder phenotype of *Sqor*^{ΔN/ΔN} compared with embryonic lethality of simple SQOR knock-out mice.

4) On page 12, the authors report that mitochondrial SQOR prevents accumulation of sulfide and persulfide not only in brain but also in heart and liver. However, no information is given on these organs before. How are SQOR levels in heart and liver? How are sulfide levels?

Please refer to our response to the Reviewer 1, Comments 2 and 3 about the SQOR levels in the heart. Information about heart SQOR was included in the second paragraph of the results and more data were added to Figures 1 and S1. We measured SQOR levels in the brain and liver and the representative immunoblots and summary graph were added as Figure S3A. We also compared ATP turnover in isolated mitochondria obtained from brain and liver and results are now presented in Figure S3, B and C. We observed that liver has approximately ~50 fold higher levels of SQOR than brain and isolated liver mitochondria are markedly resistant to inhibitory effects of sulfide on mitochondrial ATP turnover. These results further support our hypothesis that tissues with higher SQOR levels exhibit more robust resistance to sulfide-induced inhibition of mitochondrial respiration. We measured and reported brain and liver tissue sulfide levels in previous studies. We observed in mice that brain and liver tissue levels of sulfide was approximately 0.04 and 0.18 nmol/mg protein, respectively (Marutani et al., 2015; Shirozu et al., 2013). It has been reported that levels of cystathione gamma-lyase, one of the enzymes that synthesizes sulfide, are markedly higher in the liver than in the brain. Taken together it is likely that liver has higher sulfide turnover rates compared to brain.

5) the story on the ground squirrel is interesting, but I am not sure fits with the whole paper. And I am not convinced the rat is an appropriate control. In addition, why succinate accumulates in the brains of 13LG squirrels is not explained.

Reviewer's concern is well taken. However, we believe that the role of sulfide catabolism in naturally occurring hypoxia tolerance exhibited by ground squirrels supports our hypothesis that higher levels of SQOR and ability to catabolize sulfide in the brain enables hypoxia tolerance. Reviewer is correct that there is no perfect control for ground squirrels as these are wild animals with mixed genetic background. We used rats as control because they are rodents with similar body size and previous studies used rats as control of ground squirrels where hypoxia tolerance of ground squirrels was investigated (D'Alecy et al., 1990).

In regard to the observed succinate accumulation in the brains of 13LG squirrels, we added the following paragraph in Discussion:
"Breathing 5% O₂ for 5 min increased levels of sulfide, lactate, succinate, fumarate and the ratio of NADH/NAD⁺ in the brains of rats, whereas hypoxia increased only succinate in the brains of 13LG squirrels. Succinate is the universal mitochondrial feature of tissue hypoxia (Chouchani et al., 2014). The observed isolated increase of succinate confirms that the brains of 13LG squirrels are experiencing severe hypoxia, but protected from impairment of oxidative phosphorylation. Of note, knockdown of SQOR restored the ability of hypoxia to increase sulfide levels and the ratio of NADH/NAD⁺ in the brains of 13LG squirrels. These results suggest that high SQOR levels in the brain of 13LG squirrels prevent hypoxia-induced sulfide accumulation and inhibition of oxidative phosphorylation during hypoxia." (Page 32, 3rd paragraph)

Minor points

1) The order of the panels in Figure 1 is not very logical

We changed the order of panels in Figure 1.

2) Figure S1: some characters are too small (in particular the p values).

Figure S1 was divided into S1, S2, and S3 to improve its appearance and legibility.

3) Why sulfide levels are upregulated in hypoxia, as reported in the discussion, is unclear to this reviewer.

While we did not investigate the detailed mechanisms responsible for hypoxia-induced increase of sulfide levels in the current study, several mechanisms have been proposed (Arndt et al., 2017; Morikawa et al., 2012; Olson et al., 2006). Because sulfide levels are determined as a balance between sulfide synthesis and oxidation, decreased SQOR activity would increase sulfide levels. To address this hypothesis, we measured SQOR activity in the mouse brain after global ischemia induced by 2VO. We observed that 2VO decreased SQOR activity in the brain which was associated with increased sulfide levels. These results are added as Figure 7 E and discussed in a paragraph in Discussion as follows:

“There are limitations in the current study. We did not investigate how acute hypoxia increases sulfide levels in the brain. Sulfide levels are determined as a balance between sulfide synthesis and catabolism. It has been suggested that hypoxia decreases sulfide oxidation thereby increasing sulfide levels (Olson et al., 2006). Our observation that 2VO decreased SQOR activity and increased sulfide levels in mouse brain supports this hypothesis.” (Page 33, 4th paragraph)

Reviewer #3 (Remarks to the Author):

Marutani et al. report on a series of interesting and original findings indicating that levels of the mitochondrial sulfide-catabolizing enzyme sulfide:quinone oxidoreductase (SQOR) determine vulnerability of the brain to hypoxia. Starting with the discovery that H₂S preconditioning confers tolerance to hypoxia by increasing brain sulfide catabolism, they convincingly show that upregulation and mitochondrial localization of SQOR underlies this effect. Studies with isolated mitochondria further suggests that sulfides directly inhibit ETC complexes, providing a possible explanation for COX4 inhibition during hypoxia despite the high oxygen affinity of the enzyme. Other novel findings include sexual dimorphic expression of SQOR in CD1 mice that may explain greater tolerance of female mice for hypoxia. As well, the group makes a compelling argument that the well-documented hypoxia tolerance of 13-lined ground (13LG) squirrels may result from elevated SQOR levels compared to other rodents. SQOR overexpression experiments in mice and a human neuroblastoma cell line further support a neuroprotective role of sulfide catabolism. Finally, sulfide scavenging in global and focal brain ischemia models underscore the clinical relevance of the report. In light of the novelty of the findings and the comprehensiveness of the experiments, this study is likely to have a major impact in the field and to appeal to the journal's diverse readership. There are, however, some concerns that should be addressed.

1) From Fig. 3D, it is not clear whether SQOR levels are decreased in *Sqor*^{DN/DN} mice. A possible decrease in stability of the protein lacking the mitochondrial leader sequence would impact the interpretation of these experiments. Please provide a quantification.

Immunoblot analysis of whole-cell lysates of mouse embryonic fibroblasts (MEFs) established from wild-type and *Sqor*^{ΔN/ΔN} embryos was performed in three independent experiments. GAPDH was used as a control and quantified their results by calculating the band intensity of SQOR relative to GAPDH. The densitometric analysis for the western blotting shown in Fig. S6A verified no reduction of SQOR levels in the whole cell lysates of *Sqor*^{ΔN/ΔN} MEFs. This result rules out that the decrease in stability of SQORΔN protein influence the phenotypes of *Sqor*^{ΔN/ΔN} mice.

2) An obvious question from studies with the 13LG squirrel model is whether AAV-shSQOR impacts vulnerability to hypoxia. The relatively moderate effects on sulfide catabolism and NADH/NAD⁺ ratios suggest that this may not be the case.

As reviewer pointed out, AAV-mediated SQOR knockdown in 13LG squirrel brain only moderately decreased sulfide catabolism and NADH/NAD⁺ ratio. We suspect this is due to the relatively limited transfection efficiency of the AAV-based gene transfer technique in the adult brain. Please note that AAV-shSQOR injection in neonatal mice produced more robust knockdown effects than what observed in adult 13LG squirrels. Therefore, at this presumably limited efficiency of SQOR knockdown, we agree that SQOR knockdown is unlikely to impact vulnerability to hypoxia in 13LG squirrels. To improve the efficiency of transfection, AAV-shSQOR injection into the brains of neonatal 13LG squirrels may be needed. Unfortunately, such experiments have not been possible due to the travel restriction imposed by COVID-19.

3) What explains the increased persulfide levels in SPC mice breathing 21% O₂ compared to control mice (Fig. 1J)?

Increased SQOR levels induced by SPC is expected to increase oxidation of sulfide to persulfide at baseline condition. Similar increase of intracellular persulfide was observed in SH-SY5Y cells overexpressing SQOR incubated in 21% oxygen (Fig. 5B).

4) The concluding sentence in l. 157,158 is too strongly worded this early in the manuscript.

We modified the sentence as follows:

“Thus, sulfide pre-conditioning may induce tolerance to the inhibitory effects of H₂S on metabolism by upregulating sulfide catabolism in the brain.” (Page 6, 1st paragraph)

5) L. 193 (“Chronic exposure ..”) is missing references.

A reference (Weil, J.V., Jamieson, G., Brown, D.W., and Grover, R.F. (1968). The red cell mass-arterial oxygen relationship in normal man. Application to patients with chronic obstructive airway disease. *J Clin Invest* 47, 1627-1639.) was added as requested.

References

- Arndt, S., Baeza-Garza, C.D., Logan, A., Rosa, T., Wedmann, R., Prime, T.A., Martin, J.L., Saeb-Parsy, K., Krieg, T., Filipovic, M.R., *et al.* (2017). Assessment of H₂S in vivo Using the Newly Developed Mitochondria-Targeted Mass Spectrometry Probe MitoA. *J Biol Chem.*
- Blackstone, E., Morrison, M., and Roth, M.B. (2005). H₂S Induces a Suspended Animation-Like State in Mice. *Science* 308, 518.
- Budde, M.W., and Roth, M.B. (2010). Hydrogen sulfide increases hypoxia-inducible factor-1 activity independently of von Hippel-Lindau tumor suppressor-1 in *C. elegans*. *Molecular biology of the cell* 21, 212-217.
- Chouchani, E.T., Pell, V.R., Gaude, E., Aksentijevic, D., Sundier, S.Y., Robb, E.L., Logan, A., Nadtochiy, S.M., Ord, E.N., Smith, A.C., *et al.* (2014). Ischaemic accumulation of succinate controls reperfusion injury through mitochondrial ROS. *Nature* 515, 431-435.
- D'Alecy, L.G., Lundy, E.F., Kluger, M.J., Harker, C.T., LeMay, D.R., and Schlafer, M. (1990). Beta-hydroxybutyrate and response to hypoxia in the ground squirrel, *Spermophilus tridecemlineatus*. *Comparative biochemistry and physiology B, Comparative biochemistry* 96, 189-193.
- Desland, F.A., Afzal, A., Warraich, Z., and Mocco, J. (2014). Manual versus Automated Rodent Behavioral Assessment: Comparing Efficacy and Ease of Bederson and Garcia Neurological Deficit Scores to an Open Field Video-Tracking System. *Journal of Central Nervous System Disease* 6, JCNSD.S13194.
- Ikeda, K., Liu, X., Kida, K., Marutani, E., Hirai, S., Sakaguchi, M., Andersen, L.W., Bagchi, A., Cocchi, M.N., Berg, K.M., *et al.* (2016). Thiamine as a neuroprotective agent after cardiac arrest. *Resuscitation* 105, 138-144.
- Kai, S., Tanaka, T., Daijo, H., Harada, H., Kishimoto, S., Suzuki, K., Takabuchi, S., Takenaga, K., Fukuda, K., and Hirota, K. (2012). Hydrogen sulfide inhibits hypoxia- but not anoxia-induced hypoxia-inducible factor 1 activation in a von hippel-lindau- and mitochondria-dependent manner. *Antioxid Redox Signal* 16, 203-216.
- Kida, K., Marutani, E., Nguyen, R.K., and Ichinose, F. (2014). Inhaled hydrogen sulfide prevents neuropathic pain after peripheral nerve injury in mice. *Nitric Oxide.*
- Marutani, E., Yamada, M., Ida, T., Tokuda, K., Ikeda, K., Kai, S., Shirozu, K., Hayashida, K., Kosugi, S., Hanaoka, K., *et al.* (2015). Thiosulfate Mediates Cytoprotective Effects of Hydrogen Sulfide Against Neuronal Ischemia. *J Am Heart Assoc* 4.
- Morikawa, T., Kajimura, M., Nakamura, T., Hishiki, T., Nakanishi, T., Yukutake, Y., Nagahata, Y., Ishikawa, M., Hattori, K., Takenouchi, T., *et al.* (2012). Hypoxic regulation of the cerebral microcirculation is mediated by a carbon monoxide-sensitive hydrogen sulfide pathway. *Proc Natl Acad Sci U S A* 109, 1293-1298.
- Olson, K.R., Dombkowski, R.A., Russell, M.J., Doellman, M.M., Head, S.K., Whitfield, N.L., and Madden, J.A. (2006). Hydrogen sulfide as an oxygen sensor/transducer in vertebrate hypoxic vasoconstriction and hypoxic vasodilation. *J Exp Biol* 209, 4011-4023.
- Park, T.J., Reznick, J., Peterson, B.L., Blass, G., Omerbasic, D., Bennett, N.C., Kuich, P., Zasada, C., Browe, B.M., Hamann, W., *et al.* (2017). Fructose-driven glycolysis supports anoxia resistance in the naked mole-rat. *Science* 356, 307-311.

Shirozu, K., Tokuda, K., Marutani, E., Lefer, D., Wang, R., and Ichinose, F. (2013). Cystathionine gamma-lyase deficiency protects mice from galactosamine/lipopolysaccharide-induced acute liver failure. *Antioxid Redox Signal*.

Shvedova, M., Litvak, M.M., Roberts, J.D., Jr., Fukumura, D., Suzuki, T., Sencan, I., Li, G., Reventun, P., Buys, E.S., Kim, H.H., *et al.* (2019). cGMP-dependent protein kinase I in vascular smooth muscle cells improves ischemic stroke outcome in mice. *J Cereb Blood Flow Metab* *39*, 2379-2391.

Stobdan, T., Zhou, D., Williams, A.T., Cabrales, P., and Haddad, G.G. (2018). Cardiac-specific knockout and pharmacological inhibition of Endothelin receptor type B lead to cardiac resistance to extreme hypoxia. *J Mol Med (Berl)* *96*, 975-982.

Szabo, C., Ransy, C., Modis, K., Andriamihaja, M., Murghes, B., Coletta, C., Olah, G., Yanagi, K., and Bouillaud, F. (2014). Regulation of mitochondrial bioenergetic function by hydrogen sulfide. Part I. Biochemical and physiological mechanisms. *Br J Pharmacol* *171*, 2099-2122.

Volpato, G.P., Searles, R., Yu, B., Scherrer-Crosbie, M., Bloch, K.D., Ichinose, F., and Zapol, W.M. (2008). Inhaled hydrogen sulfide: a rapidly reversible inhibitor of cardiac and metabolic function in the mouse. *Anesthesiology* *108*, 659-668.

REVIEWER COMMENTS

Reviewer #1 (Remarks to the Author):

The authors have addressed all my concerns and I am happy to see the manuscript accepted for publication.

Reviewer #2 (Remarks to the Author):

The authors made a big effort to clarify their findings and addressed part of my concerns, although I am not fully convinced of their explanation on the use of NaS as a substitute for hypoxia, as there is no proof that the two responses overlap. However, the manuscript is very interesting and worth of publication.

Reviewer #3 (Remarks to the Author):

My (relatively minor) concerns have been adequately addressed. In my opinion, the manuscript is now acceptable for publication.

Stefan Strack, Ph.D.

John Paul Long Professor & Associate Chair
Department of Neuroscience & Pharmacology
University of Iowa

REVIEWERS' COMMENTS

Reviewer #1 (Remarks to the Author):

The authors have addressed all my concerns and I am happy to see the manuscript accepted for publication.

We appreciate your constructive comments.

Reviewer #2 (Remarks to the Author):

The authors made a big effort to clarify their findings and addressed part of my concerns, although I am not fully convinced of their explanation on the use of Na₂S as a substitute for hypoxia, as there is no proof that the two responses overlap. However, the manuscript is very interesting and worth of publication.

Reviewer's concern about the use of Na₂S as a substitute for hypoxia in experiments in isolated mitochondria is well-taken. To acknowledge this issue, we added sentences to the limitation section of the Discussion as follows:

"While we used a sulfide donor Na₂S as a "substitute" for hypoxia in experiments in isolated mitochondria, response to sulfide and hypoxia may not overlap. Effects of SQOR in mitochondrial energy homeostasis remains to be determined under hypoxic condition."

Reviewer #3 (Remarks to the Author):

My (relatively minor) concerns have been adequately addressed. In my opinion, the manuscript is now acceptable for publication.

Thank you very much for your constructive and encouraging comments.